

# The BRIDGE HadCM3 family of climate models: HadCM3@Bristol v1.0

Paul J. Valdes[1], Edward Armstrong[1], Marcus P. S. Badger[1,2], Catherine D. Bradshaw[3], Fran Bragg[1], Taraka Davies-Barnard[1], Jonathan J. Day[4], Alex Farnsworth[1], Peter O. Hopcroft[1], Alan T. Kennedy[1], Natalie S. Lord[1], Dan J. Lunt[1], Alice Marzocchi[5], Louise M. Parry[1,6], William H. G. Roberts[1], Emma J. Stone[1], Gregory J. L. Tourte[1], and Jonny H. T. Williams[7]

[1]School of Geographical Sciences, University of Bristol, Bristol, UK
[2]School of Environment, Earth and Ecosystem Sciences, The Open University, Milton Keynes, UK
[3]Applied Science group, Met Office Hadley Centre, Exeter, UK
[4]Department of Meteorology, University of Reading, Reading, UK
[5]Department of the Geophysical Sciences, The University of Chicago, Chicago, IL, USA
[6]Scottish Environment Protection Agency (SEPA), Perth, UK
[7]National Institute of Water and Atmospheric Research (NIWA), Wellington, New Zealand

*Correspondence to:* Paul J. Valdes (p.j.valdes@bristol.ac.uk)

**Abstract.**

Understanding natural and anthropogenic climate change processes involves using computational models that represent the main components of the Earth system: the atmosphere, ocean, sea-ice and land surface. These models have become increasingly computationally expensive as resolution is increased and more complex process representations are included. However, to gain robust insight into how climate may respond to a given forcing, and to meaningfully quantify the associated uncertainty, it is often required to use either or both of ensemble approaches and very long integrations. For this reason, more computationally efficient models can be very valuable tools. Here we provide a comprehensive overview of the suite of climate models based around the coupled general circulation model HadCM3. This model was originally developed at the UK Met Office and has been heavily used during the last 15 years for a range of future (and past) climate change studies but is now largely being replaced by more recent models. However, it continues to be extensively used by the BRIDGE (Bristol Research Initiative for the Dynamic Global Environment) research group at the University of Bristol and elsewhere. Over time, adaptations have been made to the base HadCM3 model. These adaptations mean that the original documentation is not entirely representative, and several other configurations are in use which now differ from the originally described model versions. We therefore describe the key features of a number of configurations of the HadCM3 climate model family, including the atmosphere-only model (HadAM3), the coupled model with a low resolution ocean (HadCM3L), the high resolution atmosphere only model (HadAM3H), the regional model (HadRM3) and a fast coupled model (FAMOUS), which together make up HadCM3@Bristol version 1.0. These also include three versions of the land surface scheme. By comparing with observational datasets, we show that these models produce a good representation of many aspects of the climate system, including the land and sea surface temperatures, precipitation, ocean circulation and vegetation. This evaluation, combined with the relatively fast computational





speed (up to 2000× faster than some CMIP6 models), motivates continued development and scientific use of the HadCM3 family of coupled climate models, particularly for quantifying uncertainty and for long multi-millennial scale simulations.

## 1  Introduction

This paper describes the variants of the HadCM3 family of climate models, produced by the UK Hadley Centre/Meteorological
Office, and which remain in regular use by the Bristol Research Initiative for the Dynamic Global Environment group (BRIDGE, http://www.bristol.ac.uk/geography/research/bridge). HadCM3 originated in the late 1990s with developments to the atmosphere model, HadAM3 (Pope et al., 2000), followed by developments to the ocean model (Gordon et al., 2000). The resulting coupled model, HadCM3, was one of the first models that did not require flux correction to maintain a reasonable present-day climate. It was extensively used for scientific studies of future climate change and is heavily cited, including in the 2007 IPCC
report (Solomon et al., 2007) and was still included in the 2013 report, though. The family of models have the advantage of now being very well-known in terms of their strengths and weaknesses, as numerous studies have shown and classified model biases and forecast skill at representing the mean climate state as well as variability (e.g., Toniazzo et al., 2007; Spencer et al., 2007). The model family has now been superseded by the HadGEM2 (HadGEM2 Development Team, 2011) and HadGEM3 (Williams et al., 2015) families of models.

Compared to more recent models, HadCM3 is relatively low resolution but continues to perform reasonably well, at least with respect to its mean climate (Flato et al., 2013). It also has the great benefit of computational speed, being more than 2000× faster than some versions of UKESM. This computational speed is particularly valuable for long-term simulations (necessary for many palaeoclimate simulations, studies which investigate the carbon cycle and the evolution of ice sheets) and for large ensembles (necessary for investigating the model's sensitivity to multiple parameters and quantifying the uncertainty in the
model's response to forcing).

Palaeoclimate simulations typically need many hundreds of model years to reach near-equilibrium in the surface and intermediate ocean and many thousands of years to reach equilibrium in the deep ocean. Moreover, there recently has been an increasing need to be able to consider the transient behaviour of past climate change. This has previously been tackled using the HadCM3 family of models by using either multiple "snapshot" simulations (e.g., Singarayer and Valdes, 2010; Lunt et al.,
2016; Marzocchi et al., 2015b) or by performing fully transient simulations for multi-millennial time scales (e.g., Hopcroft et al., 2014).

Faster models are also invaluable for investigating the sensitivity and robustness of results to changes in the initial and boundary conditions of the model as numerous simulations can be performed (Roberts and Valdes, 2016; Roberts et al., in review). Additionally, they are ideal for multiple runs to explore the sensitivity to a range of parameter values (e.g., using
HadCM3 variants, Davies-Barnard et al., 2014; Armstrong et al., 2016); and for performing perturbed parameter ensembles to rigorously calculate the probability density functions of either the mean or extreme climates (e.g., using HadCM3 variants, Murphy et al., 2007; Booth et al., 2012; Jackson and Vellinga, 2013; Schaller et al., 2016). Computational speed also aids more





speculative studies. For instance, many early geoengineering simulations were run using variants of the HadCM3 family of models (such as Ridgwell et al., 2009; Singarayer et al., 2009; Lunt et al., 2008; Irvine et al., 2010).

A response to the need for fast models was the development of Earth System Models of Intermediate Complexity (EMICs) (Claussen et al., 2002). These models frequently achieve their speed by heavily parameterising the atmospheric response,
even though atmospheric processes transport two-thirds of the total heat from equator to pole and play a vital role in the hydrological cycle. It is, therefore, also important to have a class of fast models that is equivalent to full atmosphere-ocean General Circulation Models (GCMs). Some EMICS do represent the dynamics of the atmosphere, for instance LOVECLIM (Goosse et al., 2010) uses a 3-level quasi-geostrophic atmosphere. Similarly, FAMOUS (part of the HadCM3 family) includes a full primitive equation atmosphere but at low resolution (Smith et al., 2008). Hence the division between EMICs and full
complexity models is becoming increasingly blurred and we consider that the HadCM3 family provides a further bridge in the spectrum of models between intermediate complexity models and full complexity, state-of-the art models.

Since its introduction, HadCM3 (and related models) has undergone a substantial number of changes, bug fixes and adaptations, such that few of the versions of the model used now are truly identical to their original model description in Pope et al. (2000) and Gordon et al. (2000).HadCM3 (and family) is still extensively used by many groups. Some groups have largely
stuck to the standard release of the model (e.g., Stainforth et al., 2005). Other groups have incorporated a variety of bug fixes and changes (in particular, many papers have used a revised land surface scheme, MOSES2.1 (e.g., Dolan et al., 2015) but this in relatively poorly documented. Therefore in this paper, we aim to rectify this for the wide range of HadCM3 variants currently in use within the BRIDGE modelling group. Our implementations of the models have diverged from other versions and so here we aim to provide clear documentation of our version of each model. We have followed a specific modelling phi-
losophy in which we attempt to minimise the differences between model configurations, particularly when changing resolution. For instance, previously published descriptions of HadRM3 and HadAM3H (Hudson and Jones, 2002) use slightly improved physics to HadCM3 but we choose to keep the same physics (except for specific changes related to resolution).

We include detailed descriptions of each module of the models, differences between variants and comparison with observations across a range of metrics. This will increase transparency, traceability, and scientific openness. By detailing the changes
and variations of these models, and providing an extensive comparison to observational data, we hope to show that these models remain useful tools for climate simulation and are suitable for further scientific use. Furthermore, we shall show that despite their relative simplicity, the models simulate the modern climate with comparable accuracy to many of the latest CMIP5 models.

To this end, we first describe the "base" model, HadCM3-M1, to which all the other models will be compared (Sect. 2). Then
we discuss different land surface schemes (Sect. 3), followed by model variants with different ocean or atmospheric resolutions (Sect. 4). Finally, we evaluate the models' performance when compared to observations and CMIP5 models, to show that they recreate many key aspects of the climate system, and show which models are more suitable for certain applications (Sect. 5).





## 1.1 Overview of HadCM3@Bristol

The family of models has at its core HadCM3. From this core, variants are derived according to resolution, land-surface scheme, and components. We choose to split the family into groups: HadCM3, HadCM3L (HadCM3 but lower ocean resolution), HadAM3 (HadCM3 but atmosphere-only), HadAM3H (HadAM3 but higher resolution), HadRM3 (HadAM3 but regional),

and FAMOUS (HadCM3L but lower atmosphere resolution).

FAMOUS is a low resolution model derived from HadCM3, sharing much of the same physics, but with some numerical modifications suitable for the low resolution and which give quicker run times. It is well documented elsewhere (Jones, 2003; Smith et al., 2008) and will not be described again in detail here, although some comparisons with FAMOUS are included for completeness.

The nomenclature adopted for the HadCM3@Bristol model variants is Had$\langle Com \rangle$M3$\langle Res \rangle$-$\langle Land \rangle\langle Veg \rangle$, where:

$\langle \boldsymbol{Com} \rangle$ (components) is one of:

    **A** – atmosphere-only model

    **C** – coupled model

    **R** – regional model

$\langle \boldsymbol{Res} \rangle$ (resolution) is one of:

    **L** – lower than standard resolution

    **H** – higher than standard resolution

    *blank* – standard resolution

$\langle \boldsymbol{Land} \rangle$ (land surface scheme) is one of:

**M1** – MOSES1 land surface exchange scheme

    **M2.1** – MOSES2.1 land surface exchange scheme

    **M2.2** – MOSES2.2 land surface exchange scheme

$\langle \boldsymbol{Veg} \rangle$ (vegetation) is one of:

    *blank* – no change to vegetation (i.e., static vegetation distribution)

**E** – vegetation predicted using TRIFFID, but in "equilibrium" mode

    **D** – same as E above, but fully dynamic model

As such, the original model described in Gordon et al. (2000), would be named HadCM3-M1 (although note that our version differs in several aspects from that in Gordon et al., see Sect. 2).



**Table 1.** Summary of the key differences between model variants. For further details of these differences and description of the features common to all variants, see the relevant sections of the text. Note that HadAM3 is identical to the atmosphere of HadCM3.

| Item | HadCM3 | HadCM3L | FAMOUS | HadAM3 | HadAM3H | HadRM3 |
|---|---|---|---|---|---|---|
| **Atmosphere** | | | | | | |
| Horizontal Resolution (n) | 96×73 | 96×73 | 48×37 | 96×73 | 288×217 | Varies with selected region |
| Horizontal Resolution (deg) | $3.75° ×2.5°$ | $3.75° ×2.5°$ | $7.5° ×5°$ | $3.75° ×2.5°$ | $1.25° ×0.83°$ | $0.4425° ×0.4425°$ or $0.22° ×0.22°$ |
| Vertical Resolution | 19 levels | 19 levels | 11 levels | 19 levels | 30 levels | 19 levels |
| Timestep (mins) | 30 | 30 | 60 | 30 | 10 | 5 or 2 |
| Dynamics sweeps/physics timestep | 1 | 1 | 1 or 2 | 1 | 2 | 1 |
| Max wind test for half timestep dynamics ($m\,s^{-1}$) | — | — | — | — | 240 | — |
| Convective precipitation grid box fraction (conv_eps) | 0.3 | 0.3 | 0.3 | 0.3 | 0.3 | 0.65 or 1.0 |
| Large scale precipitation grid box fraction (ls_eps) | 1.0 | 1.0 | 1.0 | 1.0 | 1.0 | 0.75 or 1.0 |
| Boundary layer top and number of levels (eta/level) | 0.835 / 5 | 0.835 / 5 | 0.9 / 3 | 0.835 / 5 | 0.8 / 6 | 0.835 / 5 |
| Cloud levels (eta/level) | 0.02 / 18 | 0.02 / 18 | 0.125 / 10 | 0.02 / 18 | 0.02 / 29 | 0.02 / 18 |
| Pure pressure level start (eta/level) | 0.04 / 17 | 0.04 / 17 | 0.06 / 11 | 0.04 / 17 | 0.04 / 28 | 0.04 / 17 |
| Gravity wave drag start (eta/level) | 0.956 / 3 | 0.956 / 3 | 0.9 / 3 | 0.956 / 3 | 0.956 / 3 | 0.956 / 3 |
| Surface gravity wave constant | $2.0 \times 10^4$ | $2.0 \times 10^4$ | $2.0 \times 10^4$ | $2.0 \times 10^4$ | $1.6 \times 10^4$ | $2.0 \times 10^4$ |
| Trapped lee wave constant | $3.0 \times 10^5$ | $3.0 \times 10^5$ | $3.0 \times 10^5$ | $3.0 \times 10^5$ | $2.4 \times 10^5$ | $3.0 \times 10^5$ |
| Filtering safety multiplying factor | 0.01 | 0.01 | 0.011 | 0.01 | 0.1 | — |
| Filtering wave numbers checked every | 1 timestep | 1 timestep | 1 timestep | 1 timestep | 6 hours | — |
| Steep slope horizontal diffusion off until pressure level (kPa) | 20 | 20 | 20 | 20 | 20 | 50 |
| Diffusion coefficient [*] | $5.47 \times 10^8$ | $5.47 \times 10^8$ | $4.19 \times 10^9$ | $5.47 \times 10^8$ | $4.0 \times 10^7$ | $1.7 \times 10^7$ |
| Diffusion power [*] | 3 | 3 | 4 | 3 | 2 | 2 |
| Humidity diffusion coefficient [*] | $5.47 \times 10^8$ $1.5 \times 10^8$ | $5.47 \times 10^8$ $1.5 \times 10^8$ | $2.4 \times 10^8$ | $5.47 \times 10^8$ $1.5 \times 10^8$ | $2.0 \times 10^7$ $4.0 \times 10^7$ | $1.7 \times 10^7$ |
| Humidity diffusion power [*] | 3 2 | 3 2 | 2 | 3 2 | 2 | 2 |
| RHcrit [*] | 0.95 0.7 | 0.95 0.7 | 0.91 0.687 | 0.95 0.7 | 0.95 0.8 | 0.91 0.84 0.95 |
| **Ocean** | | | | | | |
| Horizontal Resolution (n) | 288×144 | 96×73 | 96×73 | — | — | — |
| Horizontal Resolution (deg) | $1.25° ×1.25°$ | $3.75° ×2.5°$ | $3.75° ×2.5°$ | — | — | — |
| Vertical Resolution | 20 levels to 5500 m | 20 levels to 5500 m | 20 levels to 5500 m | — | — | — |
| North Atlantic Bathymetry | Standard Met. Office | No Iceland | No Iceland | — | — | — |
| Vertical Diffusion | Richardson Number dependence | Constant background mixing rate | Constant background mixing rate | — | — | — |
| Coefficient for Solar Penetration | 0 | $3.8 \times 10^{-1}$ | $3.8 \times 10^{-1}$ | — | — | — |
| Horizontal Momentum Diffusion Coefficient | $3 \times 10^3$ | $1.5 \times 10^5$ | $1.5 \times 10^5$ | — | — | — |
| Horizontal Tracer Diffusion Coefficients | Visbeck et al. (1997) latitudinally varying scheme | Constant values | Constant values | — | — | — |
| Sea Ice Diffusion | $6.7 \times 10^2$ | $2.0 \times 10^3$ | $2.0 \times 10^3$ | — | — | — |

[*] Level dependent parameters (where multiple values are given, this indicates the range from surface to top-of-atmosphere)

## 2  HadCM3-M1

This section describes the "core" model, HadCM3-M1, to which all other variants will be compared in this paper. This variant of the family was originally the most commonly used, and is still used for studies where the vegetation is known, and as such



**Table 2.** Availability of alternative land surface schemes.

| Item | HadCM3 | HadCM3L | FAMOUS | HadAM3 | HadAM3H | HadRM3 |
|---|---|---|---|---|---|---|
| MOSES1 | HadCM3-M1 | HadCM3L-M1 | FAMOUS-M1 | HadAM3-M1 | HadAM3H-M1 | HadRM3-M1 |
| MOSES2.1 | HadCM3-M2.1 | HadCM3L-M2.1 | — | HadAM3-M2.1 | HadAM3H-M2.1 | HadRM3-M2.1 |
| MOSES2.1 TRIFFID (D and E) | HadCM3-M2.1 | HadCM3L-M2.1 | — | HadAM3-M2.1 | HadAM3H-M2.1 | — |
| MOSES2.2 | HadCM3-M2.2 | HadCM3L-M2.2 | FAMOUS-M2.2 | — | — | — |
| MOSES2.2 TRIFFID (D and E) | HadCM3-M2.2 | HadCM3L-M2.2 | FAMOUS-M2.2 | — | — | — |

can be prescribed, where relatively short simulations are sufficient for the science questions being addressed, and where the ocean plays a critical role and as such high resolution is desirable (e.g., Bragg et al., 2012).

This model is a three dimensional, fully dynamic, coupled atmosphere-ocean global climate model without flux adjustment. Our version of the model is very similar to that described by Gordon et al. (2000). Our aim is to provide a brief description of this core model, and a full description of how our version differs from that in Gordon et al. (2000). A full description of the Gordon et al. (2000) version can be found in the UK Met Office technical notes http://cms.ncas.ac.uk/wiki/Docs/MetOfficeDocs; the model code is currently available to view at http://cms.ncas.ac.uk/code_browsers/UM4.5/UMbrowser/index.html.

## 2.1 Atmosphere (HadAM3)

The atmosphere component of HadCM3 is almost identical to the atmosphere component of HadAM3, which is the atmosphere-only variant with fixed sea surface temperatures (SST). HadAM3 has a Cartesian grid with a horizontal resolution of 96×73 grid points (3.75° longitude×2.5° latitude) with 19 hybrid levels (sigma levels near the surface, changing smoothly to pressure levels near the top of the atmosphere) in the vertical (Simmons and Strüfing, 1983) and uses a 30-minute timestep. HadAM3 solves the primitive equation set of White and Bromley (1995) which includes certain terms necessary to conserve both energy and angular momentum. Equations are solved through the use of a grid-point scheme, specifically the Arakawa staggered B-grid (Arakawa and Lamb, 1977), on a regular latitude-longitude grid in the horizontal. A split-explicit time scheme conserves mass, mass weighted potential temperature, moisture and angular momentum, and ensures the reliability for solving equations on long time scales, which is particularly important for climate modelling (Van der Wal, 1998).

As with any climate model, a number of parameterisation schemes are needed within HadAM3 to represent certain physical processes which occur on sub-grid scales:

– Precipitation is dealt with in two schemes: i) the large-scale precipitation scheme, and ii) the convection scheme. The large-scale precipitation scheme removes cloud water resolved on the grid-scale, i.e., frontal precipitation. This is done via a simple bulk parameterisation scheme converting water content into precipitation (Wilson, 1998). The convection scheme (Gregory et al., 1997) uses a mass-flux scheme with the addition of convective downdrafts.

– A first order scheme for turbulent vertical mixing of momentum and thermodynamic quantities is used within the boundary layer, which can occupy up to the first five layers of the model. Sub-gridscale gravity wave and orographic drag




parameterisations include the impact of orographic variance anisotropy (Gregory et al., 1998). The scheme comprises four elements: i) "Triggering" which determines whether the physical conditions within the grid-box constitute convection taking place; ii) "Cloudbase closure" which determines the intensity of convection which is determined by the mass transported through the cloudbase; iii) a transport model where temperature, moisture, wind fields and thus precipitation, are determined and iv) "Convective cloud scheme" where cloud fractions derived from convection are calculated which will be used by the radiation scheme (Grant, 1998).

- In the real world, clouds are formed on scales far below that of the coarse grid used in HadAM3, therefore there is the need for a statistical parameterisation of this variable. Probability Density Functions are used on the total water content over the grid-box mean to parameterise cloud amount/distribution and longevity (Bushell, 1998). Clouds are modelled as either water, ice or mixed-phase when the temperature in the model level is between 0 °C and −9 °C. Clouds form when the mean plus the standard deviation of the grid-cell moisture content exceeds a threshold of relative humidity (see Table 1 for numerical values). This cloud water content can then be used to produce a cloud fraction for each grid-box (Bushell, 1998). The threshold of total water content for precipitation to occur varies between land and ocean cells to account for the different levels of available cloud condensation nuclei. The scheme uses temperature through the vertical levels to determine the ice and water phases to determine cloud water content.

- Radiation is represented using the radiation scheme of Edwards and Slingo (1996). This scheme has six short-wave and eight long-wave bands and represents the effects of water vapour, carbon dioxide, ozone and minor trace gases. A background aerosol climatology following Cusack et al. (1998) increases the atmospheric absorption of short-wave radiation relative to previous versions representing a significant improvement. The 3A longwave and shortwave spectral scheme is used and is an improvement over the previous versions as it allows the freedom of choices of cloud parameterisation, gases and aerosols to be included through spectral input files (Edwards, 1998).

Boundary conditions for the model include the land sea mask, orography and its subgrid scale variability (originally derived from the US Navy updates 10' dataset), and a range of soil and vegetation parameters (originally derived from data in Wilson and Henderson-Sellers (1985). The model also needs to be initialised with soil moisture and snow cover (based on Willmott et al., 1985), and deep soil temperatures (empirically derived using Warrilow et al. (1986). When the model is run in atmosphere-only mode, i.e., HadAM3, sea surface temperature and sea ice (concentration and depth) are required to be prescribed.

## 2.2 Ocean

The ocean component has a horizontal resolution of $288 \times 144$ grid points ($1.25° \times 1.25°$) (Gordon et al., 2000). Therefore six ocean grid cells correspond to each atmosphere grid cell. In order to simplify the coupling of the atmosphere and ocean models, the land-sea mask is defined at the atmosphere resolution; therefore, the ocean model's coastlines appear relatively coarse. In the vertical there are 20 depth levels with finer definition at the ocean surface, with the top-most model layer being 10 m thick



and the bottom-most 616 m thick. The ocean timestep is one hour. The ocean and atmosphere modules are coupled once a day with no flux adjustment necessary.

The ocean model is based on the model of Cox (1984) and is a full primitive equation, three dimensional model of the ocean. A second order numerical scheme is used along with centred advection to remove nonlinear instabilities. The Arakawa B-grid is used for staggering of tracer and velocity variables, allowing for more accurate numerical calculations of geostrophically balanced motion. It uses a rigid lid which eliminates fast external mode gravity waves found in the real ocean, thus allowing for longer timesteps, and with the result that there is no variation in the volume of the ocean.

As with the atmosphere, the ocean model also requires a number of parameterisations:

- The ocean mixed layer is represented by the Kraus and Turner (1967) model which assigns 15 % of gravitational potential energy and 70 % of wind-stress energy to turbulent kinetic energy, which is mixed out exponentially with depth. At all depths, five iterations of convective mixing are carried out at each timestep. Tracer and momentum mixing is modelled using the K-Theory scheme. Within the mixed layer a simplified version of the Large et al. (1994) scheme is employed: below this the Pacanowski and Philander (1981) K-Theory parameterisation is used.

- Further vertical mixing is provided at all depths as the sum of depth-dependent constant background term and a term dependent on Richardson number (Pacanowski and Philander, 1981).

- Momentum mixing is approximated using diffusion that is governed by a coefficient that consists of two terms: a constant background value and a term dependent on the local Richardson number. For tracers, diffusion increases with depth as detailed in Table A of Gordon et al. (2000).

- Horizontal eddy mixing of tracers is carried out using the isopycnal parameterisation of Gent and McWilliams (1990), with thickness diffusion coefficients modified following the method of Visbeck et al. (1997). Isopycnal mixing uses the Griffies et al. (1998) implementation of the Griffies et al. (1982) scheme. The along-isopycnal diffusion coefficient is 1000 $m^2\,s^{-1}$. Horizontal mixing of momentum is performed using a latitudinally varying formulation which, coupled with the finer resolution of the ocean grid, enables western boundary currents to be resolved.

- There is no direct connection between the Mediterranean Sea and Atlantic Ocean so it is modelled as a "diffusive pipe" by completely mixing the easternmost point of the Atlantic with the westernmost point of the Mediterranean. Mixing occurs over the top 13 layers, to a depth of 1200 m, on the assumption that Mediterranean water will sink to at least this depth. A similar parameterisation is applied in the outflow of the Hudson Bay.

- Ice sheets are not modelled dynamically, therefore the snow accumulation on surface land ice points and over isolated water bodies must be balanced by loss through a notional iceberg calving that is represented as a time-invariant freshwater flux (which, because of the rigid lid, is converted to a virtual salinity flux). This is distributed around the edge of the ice sheets and polar oceans. River runoff is instantaneously transferred to the ocean using a prescribed runoff map.



The modern bathymetry for the model is derived from the ETOPO5 reconstruction (Edwards, 1989) using a simple smoothing algorithm. The geometry of some significant channels is modified from the resulting coarse interpolation to ensure a more realistic model performance (Gordon et al., 2000). For example, the Greenland–Scotland ridge and Denmark Strait have significant sub-gridscale channels which are lost in the smoothing and so have been re-created by deepening single cell width channels in three locations along the ridge to reproduce the mean outflow to match observations, and the bathymetry around Indonesia is modified to ensure that flow occurs between Indonesia and Papua New Guinea but not between Indonesia and the mainland of Asia.

## 2.3 Sea ice

Sea ice is calculated as a zero layer model on top of the ocean grid. Partial cell coverage of sea ice is possible up to 0.995 in the Arctic and 0.98 in the Antarctic according to the parameterisation of sea ice concentration due to Hibler (1979). Ice forms primarily by freezing in leads, although ice can also form from snow falling on existing ice. It is assumed to freeze at the base of the sea ice at the freezing point of $-1.8\,^\circ$C. A constant salinity is assumed for ice, with the excess salt on melting/formation added as a flux into the ocean. Sea ice dynamics are simply parameterised: the surface wind stress over sea ice is applied to the ocean beneath the ice, and the ice thickness, concentration and accumulated snow then drift following the ocean currents in the top model layer (Gordon et al., 2000). The maximum depth that sea ice can reach due to convergence from drift is limited to 4 m in depth, although it may subsequently thicken further due to freezing. The albedo of sea ice is set at 0.8 for temperatures below $-10\,^\circ$C and 0.5 for temperatures above $0\,^\circ$C, with a linear variation between these values.

## 2.4 Land Surface Scheme: MOSES1

The land surface scheme MOSES (Met Office Surface Exchange Scheme) is built upon the previous Met Office land surface scheme (UKMO) (Warrilow and Buckley, 1989). In the Gordon et al. (2000) version of HadCM3, MOSES Version 1 MOSES1, is used. A technical overview of MOSES1, a comparison to its predecessor (UKMO) and its climatological impact is provided by Cox et al. (1999).

In addition to calculating the fluxes of water and energy, MOSES1 incorporates the physiological impact of atmospheric carbon dioxide, water vapour and temperature on photosynthesis and stomatal conductance. It accounts for the effects of freezing and melting of soil moisture in four soil layers whose thicknesses are a function of the soil heat capacity and conductivity of the grid cell. Both vegetated and non-vegetated land surface types are characterised by a set of surface properties that are not updated during the model run. The canopy scheme is based on that used in Warrilow et al. (1986).

MOSES1 has two sets of prescribed land surface property attributes, which are input into the model via two external files. The soil attributes are volumetric soil moisture concentration at the wilting point, critical point, field capacity, and saturation, the saturated hydrological soil conductivity, the Clapp–Hornberger B exponent, the thermal capacity of soil, thermal conductivity of soil, and the saturated soil water suction. The vegetation attributes are root depth, snow free albedo, stomatal resistance to evaporation, surface roughness, canopy water capacity, infiltration enhancement rate, deep snow albedo, leaf area index and canopy height of vegetation fraction. All of these attributes are derived from the Wilson and Henderson-Sellers (1985) data set.

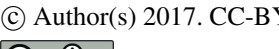



## 2.5 Modifications for the Bristol Version of HadCM3-M1

We have benchmarked the standard version of HadCM3-M1 supplied by the UM Meteorological office against existing model results from the published Hadley centre version of Gordon et al. (2000) and confirmed that we could reproduce the results within the normal statistical variability of the model. Subsequently a few relatively minor changes have been made. These

include:

- Correction of a small bug in the Visbeck scheme which was originally included in the standard configuration of the model to ensure compatibility with previous versions.

- Use of versions of the radiation and primary field advection schemes that are scientifically identical to the standard version and which make the model faster but are not bit reproducible.

- Fixes to a few array bounds errors (which may or may not have an impact on the scientific results).

- Numerous other bug fixes which did not change the science but corrected problems with some aspects of the code and diagnostic outputs.

- There were two small bugs in the conservation of atmospheric mass, and the computation of vertical velocity, fixes to which are not included in the standard release version of HadAM3 but are included in HadCM3. We include these bug-

fixes in all versions of the code so that our atmosphere model (HadAM3) is 100 % identical to the atmosphere component of our version of HadCM3.

The overall impact of these changes on the climate simulation is very small.

## 3 Alternative Land Surface Schemes

Section 3 describes the MOSES1 land-surface scheme which is used in the standard version of HadCM3. Here we describe

two other versions, MOSES2.1 and MOSES2.2, as well as the vegetation component TRIFFID.

### 3.1 MOSES2

MOSES1 requires maps of vegetation properties, such as root depth and leaf area index which have to be prescribed (normally in a set of external files), and as such is not very suitable for an interactive vegetation model. As part of the process of developing a dynamic vegetation module for HadCM3, an upgraded land surface scheme was also developed, called MOSES2. The first

version of this scheme, MOSES2.1, is the original scheme used in early work with dynamic vegetation (Cox et al., 2000). This version was originally coupled to HadCM3LC, which is a flux-corrected low resolution version of HadCM3 which includes a carbon cycle. MOSES2.1 was further developed by Michel Crucifix for use in HadCM3 as part of the Paleoclimate Modelling Intercomparison Project Phase II (PMIP2) (Braconnot et al., 2007). Subsequently, a second version of MOSES2 was developed, MOSES2.2 (Essery et al., 2001, 2003) which was similar scientifically to MOSES2.1 but had improved code structure and has





become the initial core of the land surface model JULES (Best et al., 2011; Clark et al., 2011). At the University of Bristol, we have mainly used MOSES2.1, with MOSES2.2 only being used in a few specific contexts such as for investigating changes in atmospheric chemistry (Valdes et al., 2005; Beerling et al., 2011) because it can include additional parameterisations of isoprene emissions. MOSES2.2 is also used in FAMOUS (Williams et al., 2013).

A detailed discussion of the upgrades between MOSES1 and MOSES2.2 is provided in Essery et al. (2003) and a full and complete technical overview of MOSES2.2 in Essery et al. (2001). But so far there have been no clear comparisons as to how MOSES2.2 differs scientifically or technically from MOSES2.1, despite MOSES2.1 being the core version used at Bristol. The following sections aim to rectify this and clarify the differences between MOSES2.2 and MOSES2.1, after first outlining how MOSES2.2 differs from MOSES1.

### 3.1.1   Differences between MOSES2.2 and MOSES1

Compared to MOSES1, MOSES2.2 has major upgrades to all aspects of the land surface exchange and the surface radiation scheme. The surface radiation scheme has an updated coupling between the land surface and atmosphere, including the calculation of surface net radiation and surface heat and moisture fluxes. MOSES2.2 allows fractional coverage of different surface types on a sub-grid scale. There are nine land-surface types explicitly modelled at a sub-grid scale, each with a set of charac-
teristic parameters. MOSES2.2 can be fully coupled to the dynamic vegetation model TRIFFID (see Sect. 3.2). The fractional coverage includes five plant functional types (PFTs): broadleaf trees, needleleaf trees, shrubs, C3 (temperate) grasses and C4 (tropical) grasses. Each have different values for a range of phenological characteristics including Leaf Area Index (LAI), albedo, and canopy interception. The remaining four are non-vegetated surface types; urban, inland water, bare soil, and ice. Excluding ice type, each land-surface gridbox can be made up of any mixture of the other eight surface types. Land ice has to
have fractional cover of 0 or 1 only.

Unlike MOSES1, the surface energy balance in MOSES2.2 is explicitly solved for each surface type (tiles), and then weighted by the fractional area of the surface types within the grid box. This produces the gridbox average surface temperature and soil moisture and fluxes of longwave, shortwave, sensible, latent and ground heat. In contrast, air temperature, humidity and wind speed on atmospheric levels above the surface are treated as homogeneous across the gridbox. Likewise,
soil temperatures and moisture contents below the surface are also treated as homogeneous. The aerodynamic surface roughness lengths are calculated explicitly according to the canopy height and the rate of change of roughness length with canopy height for each tile. This roughness length is used to calculate surface-atmosphere fluxes of heat, water, momentum and $CO_2$. The surface albedo determines the amount of downward shortwave heat flux that is reflected at the surface. The surface albedo for fractional covered vegetated surface types (unweighted) is described by the snow-free and cold deep snow albedos. The
soil albedo is defined according to colour and moisture content. LAI is also used in determining the surface albedo for surfaces covered by vegetation.

The hydrological cycle in MOSES2.2 is more similar to MOSES1 with small changes for the interactions with vegetation. However, it continues to treat each tile separately so extraction of water from the soil is calculated for each tile and then weighted summed to give the grid box average. Precipitation is partitioned into interception (via the canopy), throughfall, run-





off and infiltration into the ground. Different parameters apply to each vegetation type. Canopy water refers to the precipitation intercepted by plant leaves available for free evaporation. MOSES 2 uses the same four soil layers as MOSES1, with thicknesses from the surface downwards set to 0.1 m, 0.25 m, 0.65 m and 2 m. Moisture content of the upper soil layer (0.1 m) is increased via snow melt and throughfall and decreased according to evaporation from the soil layer, flow of water into lower layers and

draw up of water via plant roots. The extraction of water from any particular soil layer is proportional to the water lost by evapotranspiration reflecting the vertical distribution of roots. The five PFTs have different root depths, such that trees are able to access moisture from soil layers at deeper depths compared with grasses and shrubs. The soil moisture content and soil water phase changes and the associated latent heat describe the thermal characteristics of soil that determine, via discretised form of the heat diffusion equation, the subsurface temperatures. Subsurface soil temperatures are determined by the diffusive heat

fluxes into and out of a soil layer and the heat flux advected from the layer by the moisture flux.

MOSES2 requires similar soil parameter inputs to MOSES1 although it additionally requires bare soil albedo and soil carbon content of the soils. However, the vegetation properties are very different. MOSES1 required inputs of grid box average LAI, root depth, etc, whereas MOSES2 requires prescribed inputs of the fractional types of each surface type, the LAI and canopy height of each vegetated PFT, and the overall canopy conductance. It also includes a disturbance fraction that represents

agriculture. If using dynamic vegetation (TRIFFID), then these fields (except for disturbance) are only used for initialisation and the model will dynamically update them.

### 3.1.2   Differences between MOSES2.2 and MOSES2.1

There are a number of key differences between MOSES2.1 and MOSES2.2, and a number of smaller modifications between the versions. These major changes include:

1. MOSES2.2 uses a spectral albedo scheme to calculate separately the diffuse and direct-beam surface albedos. This scheme is not used in MOSES2.1, although modifications can be added to include it.

2. MOSES2.2 uses a spectral snow albedo model that includes a prognostic grain size that characterises the ageing of snow and its impact on snow albedo. This is not present in MOSES2.1.

3. MOSES2.2 also introduces a new calculation of evapotranspiration from soil moisture stores, as well as a different
parameterisation of bare soil evaporation

4. Supersaturation in the soil layer is treated differently in the two versions of MOSES2. In MOSES2.2, supersaturation results in an increase in surface run-off. In contrast, supersaturation in MOSES1 and MOSES2.1 is managed via an increase in downward flow into the deeper soil layers and so is removed via subsurface runoff.

Tests carried out in which MOSES2.1 is gradually changed to MOSES2.2 show that the first two changes have a particularly
big effect on surface temperature whereas the third difference substantially alters soil moisture. Supersaturation changes have a big impact on the partitioning of runoff between surface and sub-surface and also have a small impact on the soil moisture.



There are also a number of smaller changes (such as using an implicit soil moisture scheme in MOSES2.2 compared to an explicit scheme in MOSES2.1 (and MOSES1) but these do not result in a major change to the climate. MOSES2.2 also had some major restructuring of the Fortran code.

Additionally, in the default version of MOSES2.1 (used until recently), the rate of respiration increases almost exponentially with temperature (Tindall, pers.comm.). As a result, in some conditions such as during the Amazon dry season, respiration excessively increases and this decreases soil moisture which consequently inhibits tree growth. In MOSES2.2, the impact of temperature on respiration rate declines at high temperatures. This revised respiration rate reduces drying and dieback of trees. This has now become the default for the Bristol variant of HadCM3-M2.1 too.

## 3.2 TRIFFID

MOSES2.1 and MOSES2.2 both have the capacity to be run in coupled mode with a dynamic vegetation and terrestrial carbon cycle scheme, TRIFFID (Top-down Representation of Interactive Foliage and Flora Including Dynamics) (Cox et al., 1998; Cox, 2001). TRIFFID predicts the distribution and properties of global vegetation based on plant functional types using a competitive, hierarchical formulation. The performance and sensitivity of TRIFFID has been compared with a variety of other dynamic vegetation models (Sitch et al., 2008) and an updated version of TRIFFID is used in both the latest Coupled Model Intercomparison Project (CMIP5) model HadGEM2-ES (Collins et al., 2011) and in JULES (Clark et al., 2011).

In the model configurations presented here, TRIFFID is normally only used with MOSES2.1 because of a dry bias in MOSES2.2 which is manifested by an overly dry surface climate over the Eurasian continent in summer. This results in loss of vegetation if used with dynamic vegetation. TRIFFID updates the five vegetation PFTs and the bare soil fraction, all of which can change dynamically. TRIFFID can be run in two different modes:

– Equilibrium mode, where TRIFFID runs for 50 years of TRIFFID for each 5 years of the climate model run. The fluxes between the land and the atmosphere are calculated and averaged over 5 years. This is particularly valuable for quick spin-up of the vegetation and soil carbon.

– Dynamic mode, where TRIFFID is run every 10 days. Fluxes are averaged over 10 days; as such high frequency variability is accounted for. This mode is the standard for full runs of the coupled model.

MOSES2 passes the averaged fluxes of carbon to TRIFFID which calculates the growth and expansion of the existing vegetation, and updates the land surface parameters based on the new vegetation distribution and structure. TRIFFID calculates areal coverage, leaf area index (LAI) and canopy height for five defined plant functional types (PFTs): broadleaf tree, needleleaf tree, C3 grass, C4 grass and shrub. These PFTs respond differently to climate and $CO_2$ forcing (e.g., C3 and C4 grasses use different photosynthetic pathways), and also impact differently on the physical properties of the land surface, i.e., possessing different aerodynamic roughness lengths and albedo properties. Broad and Needleleaf trees and C3 and C4 grasses react independently within the model due to their unique parameter sets. C4 plants use water more efficiently than C3 plants, requiring less water to produce the same amount of biomass. Overall, C4 plants have the highest critical humidity deficit and



temperature range, meaning that in high temperature, low moisture environments they will do better than other PFTs, even though the competition model would normally favour trees.

All PFTs can co-exist within the same grid box, each possessing a fractional coverage that is equivalent to the population size. The fractional coverage co-existence approach allows smooth transitions to occur when the vegetation distribution changes rather than the sudden discontinuities that would occur in a "dominant" PFT only approach (Svirezhev, 2000). However, the Lotka–Volterra equations used in TRIFFID mean that each grid cell in the model tends to converge on one dominant plant functional type (Hughes et al., 2006). Competition is essentially based on a height hierarchy of trees > shrubs > grasses. Each terrestrial grid square has a small minimum content of each plant functional type, regardless of location and competition, as a "seeding" fraction (Cox, 2001). This ensures that no PFT can become extinct and can regenerate when conditions become appropriate. TRIFFID can specify areas of agricultural crops as C3 and C4 grasses, without competing land types (Cox, 2001).

The terrestrial Net Primary Productivity (NPP) is calculated by a coupled photosynthesis-stomatal conductance model (Cox et al., 1998). Factors affecting the rate of photosynthesis are the humidity deficit, the photochemically active radiation, soil moisture and LAI. The maximum rate of photosynthesis is directly related to the leaf temperature and the upper and lower temperatures for photosynthesis (defined individually for each PFT). Carbon is stored in the vegetation and soil stores.

The predicted vegetation in each grid box feeds back into the climate system in a number of ways, principally through evapotranspiration from the canopy, alteration of surface albedo, and through alteration of mixing at the boundary layer between the surface and the atmosphere (due to changes in roughness length).

## 4 Variants with Differing Resolution

### 4.1 HadCM3L

HadCM3L comprises the same model components as HadCM3, but with a lower resolution ocean which matches the standard atmosphere resolution of 96×73 grid points (3.75°×2.5°). It can be run with all versions of MOSES, with or without TRIFFID, in the same manner as HadCM3. We tend to use HadCM3L when long simulations are required. For instance, when the land-sea mask and/or bathymetry are substantially changed from those of modern, it can take many thousand years of intergration to get the deep ocean into equilibrium. As such, HadCM3L has been used extensively for our pre-Quaternary climate modelling work (e.g., Marzocchi et al., 2015b; Bradshaw et al., 2015; Kennedy et al., 2015; Loptson et al., 2014).

The implementation of the atmosphere and land surface schemes is identical to HadCM3. There are some differences in the ocean due to its lower resolution, some of which are substantive differences required either to maintain stability or to reproduce the present-day climate without the need for the flux corrections used in earlier versions of the model, and some of which are simple scalings of parameters to give the same scientific behaviour as HadCM3 at the lower resolution. These differences between HadCM3 and HadCM3L, which are described below, are generally consistent with work done to optimise the FAMOUS model (Jones, 2003), which has the same ocean resolution as HadCM3L. Note that the Bristol version of HadCM3L is very different from the Met Office version. The Met Office version was mainly used for the early carbon cycle





work (Cox et al., 2000) but required significant flux corrections to ensure that the Atlantic surface climate was reasonable. Our version does not require flux correction because of changes in bathymetry described below.

### 4.1.1 North Atlantic Bathymetry: "No Iceland"

As described in Sect. 2.2, care was taken when developing HadCM3 to define the bathymetry of the North Atlantic in order

to ensure that the appropriate flow through the Denmark Straits was captured. This flow is lost when the ocean resolution is reduced in HadCM3L as the channel between Iceland and Greenland becomes less than a single grid-cell wide (on the velocity grid) and thus no flow is permitted. Jones (2003) investigated potential modifications to allow increased heat transport through this region, thus alleviating the unrealistic buildup of sea ice in the Nordic Sea, and concluded that the removal of Iceland was the preferred solution. With this modification, the flux corrections required in previous ocean versions of this resolution

(HadCM2) (Johns et al., 1997) to maintain a stable meridional overturning circulation are no longer required.

This change also has a knock on effect on the land surface (and ultimately the atmosphere) in that the 2 cells that define Iceland have been removed.

### 4.1.2 Ocean Diffusion Schemes HadCM3L vs. HadCM3

**Vertical diffusion**

In HadCM3L, the Richardson Number dependence of the vertical mixing coefficient is replaced with a constant background mixing rate, as it is in FAMOUS. Jones (2003) describes problems encountered with FAMOUS in the interaction between the mixed layer and deep vertical mixing schemes, but was found to have little impact on the solution because of the relatively low resolution.

HadCM3L uses a different calculation for the density of seawater from HadCM3. HadCM3 calculates all densities relative to

a reference level at the surface using the updated equation of state for seawater of UNESCO (1981). This can result in negative density gradients in the deep ocean and hence a negative Richardson Number, which in turn can produce very high diffusivities at depth which Pacanowski and Philander (1981) was never intended to handle (Rickard, 1999). HadCM3L instead derives Bryan and Cox (1972) third order polynomials for each 250 m depth span of the ocean (Foreman, 2005) to fit the Knudsen–Ekman equation for the density of seawater and does not produce negative density gradients (Rickard, 1999), but the range of

salinities covered may be insufficient for some applications. This choice of diffusion scheme is consistent with that used in FAMOUS.

**Ocean diffusion parameters**

HadCM3L uses different coefficients for a number of aspects of the diffusion formulation, as described in Table 1. All of these values are consistent with those used in FAMOUS. In addition, the Visbeck et al. (1997) scheme for the calculation of isopycnal

thickness diffusion coefficients, introduced in HadCM3 to improve resolution of currents such as Western boundary currents




on the 1.25° grid, is not used in HadCM3L. Instead, fixed values of the coefficients for surface ocean diffusion, deep ocean diffusion and scale depth are specified, as in FAMOUS.

### 4.1.3 Solar Penetrative Radiation HadCM3L vs. HadCM3

The ratio of solar radiation components (RSOL), which is set to 0.0 in HadCM3 is set to $3.8 \times 10^{-1}$ in HadCM3L, as it is in

FAMOUS.

### 4.1.4 Islands HadCM3L vs. HadCM3

HadCM3 defines 6 islands in the barotropic solution, around which non-zero depth-integrated flow is permitted: Antarctica, Australia, New Zealand, the Caribbean, Madagascar and Iceland. In HadCM3L, there is no island for Iceland as this is entirely absent from HadCM3L and Madagascar is also not defined as an island due to its proximity to Africa.

**4.1.5 Scaling and Retuning for Lower Resolution HadCM3L vs. HadCM3**

There are a number of further parameters that are scaled or retuned appropriately for the lower resolution ocean to give the same behaviour as HadCM3. These parameters are all scaled in the same manner as for FAMOUS and described in Table 1.

### 4.2 HadAM3H

HadAM3H is a higher resolution version of the atmosphere-only variant, HadAM3. It is used for studies in which the atmo-
spheric circulation is critical, and as such is best represented at high resolution. Its horizontal resolution is three times greater than HadAM3 both latitudinally and longitudinally, i.e., 288×217 grid points (1.25°×0.83°). The number of vertical levels is increased from 19 to 30, with the extra levels being concentrated close to the Earth's surface and the upper levels remaining similar to HadAM3. The higher spatial resolution requires a smaller timestep of 10 minutes. It may be used with either the MOSES1 or MOSES2.1 land surface scheme, and can be used with TRIFFID, though this has rarely been done. The time
stepping algorithm is slightly different, in that the dynamics can be updated multiple times between the full physics time steps. In HadAM3H, we use two dynamic per physics time step to allow for improved numerical stability of the model. Various diffusion coefficients, critical relative humidity and parameters for the gravity wave drag scheme have been retuned to account for the change in resolution, as documented in Table 1. Otherwise the model has identical physics to HadAM3 and has had no further changes. This model is different to that used by the Met Office (e.g. Hudson and Jones, 2002; Arnell et al., 2003) which
keeps to 19 levels in the vertical but has some changes to the parameterisations, particularly in the boundary layer.

### 4.3 HadRM3

HadRM3 is the regional climate model (RCM) version of HadAM3 which has been used when representation of high resolution atmospheric processes is important, such as around orography or studying extreme events. It can be configured for any domain size and location has commonly been used for studies over Europe (Jones et al., 1995), the Arctic (Day et al., 2013) and





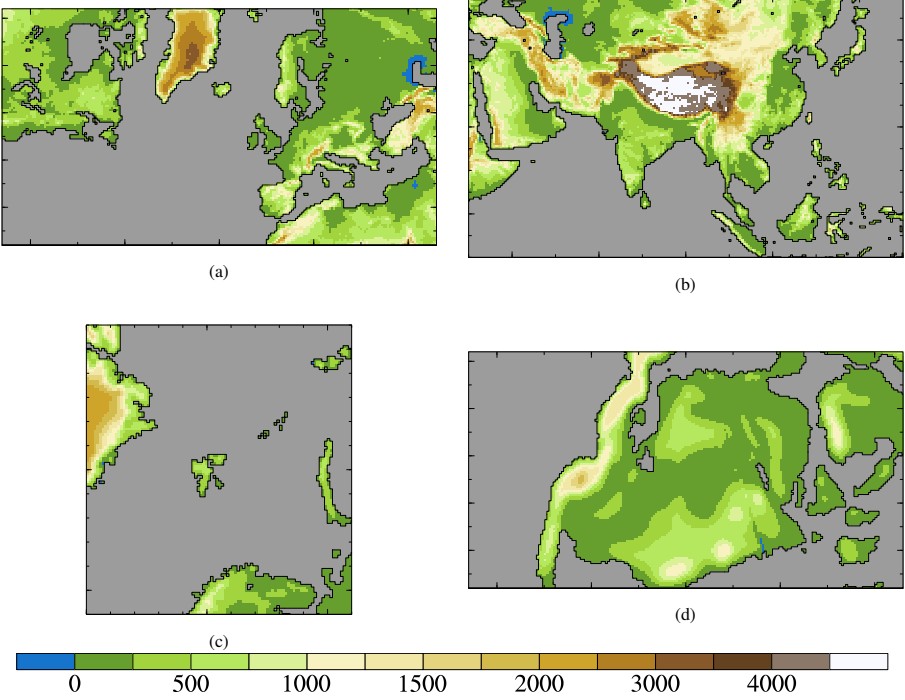

**Figure 1.** Land/sea mask and orography (sea coloured grey, land height in meters) for four configurations of HadRM3. (a) Shows the standard European domain at 0.44° resolution, (b) shows the equivalent domain for E. Asia, (c) shows a configuration for the Arctic and Svalbard at 0.22° (as used in Day et al., 2012), and (d) a N. America/European configuration for the early Cretaceous at 0.44° resolution (as used in Haywood et al., 2004)

Svalbard (Day et al., 2012), and the East Asian Monsoon region (Bhaskaran et al., 1996). It has also been used to model deep time (Haywood et al., 2004).

The BRIDGE version of HadRM3 is based on the same fundamental physics and model structure of HadAM3, and currently is only available with the MOSES1 land surface scheme. We again do not make any substantial changes to the physical parameterisations so the model is largely identical to HadAM3 except for parameters sensitive to resolution.

Regional climate models require either fixed or time evolving data on the large scale and global atmospheric and ocean response to climate forcings to be provided to them at their lateral (atmospheric) and sea surface boundaries, such as potential temperature and specific humidity. The common experiment set-up, used here, is a one-way nested approach, where no information is fed back into the GCM simulation, but the large scale atmospheric circulation patterns, such as the location of the jet streams, are fed in through the Lateral Boundary Conditions (LBCs). For a RCM to have a "parent" GCM is rare, offering a unique opportunity to investigate the effects of dynamical downscaling without modification (or contradiction) of the physics between the driving GCM and the RCM at the lateral boundaries. LBCs are updated every 6 hours and linearly interpolated for timesteps in between. A 4-grid smoothing is applied to global model data entering the regional model domain. Therefore





typically, HadRM3 has been run here using HadAM3 or HadCM3 to produce the lateral boundary conditions, sea surface temperature, and sea ice concentration data, although there have been experiments using SSTs from HadISST and HadGEM, as well as other models in the CMIP5 experiment to analyse the sensitivity of the model to its boundary conditions.

HadRM3 is run on a standard lat-long grid with the pole rotated so that the centre of the domain of interest lies across the
equator within the RCM's grid from of reference (see Fig. 1) to reduce variation in the areas of the grid cells. The timestep of the model is five minutes to maintain numerical stability with the increase in spatial resolution which is commonly $0.44° \times 0.44°$ ($\sim 50\,\text{km} \times 50\,\text{km}$) but has also been run at $0.22° \times 0.22°$ ($\sim 25\,\text{km} \times 25\,\text{km}$). Lateral boundary conditions are typically provided every 3 or 4 hours and linearly interpolated to each timestep. The main difference between HadRM3 and HadCM3/HadAM3 in terms of atmospheric dynamics is in the sub grid scale diffusion applied to the horizontal wind component to prevent the
accumulation of energy at the smallest scales and noise (see Table 1). In addition, the parameters which control the proportion of a grid box over which convective and large scale precipitation are assumed to fall, as well as diffusion parameters, vary compared to HadAM3 (see Table 1, variables `conv_eps` and `ls_eps`).

Simulations using the regional climate models have enabled improved spatial representation of temperature and precipitation patterns and response to climate forcings, particularly around mountains and coastlines. The increase in resolution also
improves the simulated temporal variability, including simulation of extremes (Durman et al., 2001).

## 5   Comparison with data

The aim of this section is to qualitatively and quantitatively evaluate the suite of HadCM3@Bristol models in terms of their ability to recreate key aspects of the climate system relative to observations, and other models within the CMIP5 family. In the following subsections, a selection of observational datasets are compared to multiple modelled climatic variables. Details
on the datasets used for each variable are briefly outlined in each subsection. This is not intended to be a complete model evaluation; however, it will highlight that some variants do a more realistic job than others at representing various environmental processes. Where appropriate, stronger or weaker models will be highlighted, and some other CMIP5 models will be shown for comparison. Because much of our work at Bristol involves carrying out palaeoclimate or idealised simulations, our standard control simulations are static pre-industrial simulations, similar to the CMIP5 DECK pre-industrial simulation (Eyring et al.,
2016a). However, most observational datasets are of the instrumental record, typically the last few decades. This is to be considered when interpreting our evaluation, although it is likely that differences between pre-industrial and the instrumental period are generally small relative to the model biases.

A quantitative evaluation (global RMSE analysis) of the four base state BRIDGE models, namely HadCM3 with the MOSES1 land surface scheme, HadCM3 with MOSES2.1, HadCM3L with MOSES2.1 and FAMOUS with MOSES1, is
performed against reanalysis and/or observational data and shown alongside new and predecessor models from the CMIP5 database (Fig. 2; BRIDGE models highlighted in red). Here we make use of the ESMValTool(v1.0); a community diagnostic and performance tool (Eyring et al., 2016b) to assess and compare the magnitude of known systematic biases inherent in all climate models. Better understanding of these biases is instrumental in diagnosing their origin and a models ability to repro-



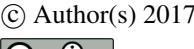

duce observed spatial and temporal variability and trends in various atmospheric (e.g., Large-scale circulation) and oceanic phenomena (e.g., ENSO). CMIP5 model data is provided from http://www.ceda.ac.uk, while observational (obs4MIPs; Ferraro et al., 2015) and re-analysis (ana4MIPs; Ferraro et al., 2015) data are provided from https://www.earthsystemcog.org, all conforming to the CMIP5 format. Here the BRIDGE models have also been standardised to the CMIP5 format. Further, models

and observations are re-gridded to the coarsest resolution within the ESMValTool framework for evaluation. Table 3 details the different metrics used for the evaluation of the historical model simulations in Fig. 2. The BRIDGE models are only pre-industrial climatologies (30-year) without any year-on-year historical forcing, however this is not expected to be detrimental for the evaluation. The results in Fig. 2 demonstrate that the BRIDGE suite of models, with the exception of FAMOUS-M1H, accurately reproduces observed global spatio-temporal patterns. Indeed, HadCM3-M1, HadCM3-M2.1 and in most respects

HadCM3L-M2.1, outperform many of the higher fidelity CMIP5 models with lower RMSE when compared to the observations, particularly in respect to global air temperature (at 850 hPa and 200 hPa), U-wind (at 850 hPa and 200 hPa) and 1.5 m surface temperature. It is likely that the course resolution of FAMOUS has a detrimental impact on its performance. The following sections provide a more detailed evaluation of various atmosphere, ocean and land surface variables in the BRIDGE model suite.

## 5.1 Atmosphere

### 5.1.1 Temperature

The land surface air temperature (SAT) observation data used here is the University of East Anglia high resolution climatology for 1960–1990 (CRU CL v2.0) (New et al., 2002). This record is based on a range of weather stations totalling more than 10 thousand stations for temperature and more than 25 thousand stations for precipitation, with the best spatial coverage over North

America, Europe and India and the sparsest spatial coverage over the interiors of South America and Africa and Antarctica. Modelled SAT fields were masked to model land points only and differences to observations were done at the same resolution as the relevant model, as shown in Fig. 3.

It should be noted that the comparison between the versions of the HadCM3 family and the observed CRU CL v2.0 data is not a "clean" comparison. The observed data is for 1960–1990 whereas all model simulations are for the pre-industrial period.

In the case of HadAM3 simulations, the SSTs used are the 1870–1900 means of HadISST. To evaluate the impact of this effect, we examined the CMIP5 historical run of the Hadley Centre version of HadCM3-M1. The differences between the 1960–1990 climate means compared to the 1860–1890 climate means were generally small compared to the model biases, with the overall mean warming between the two periods being 0.6 °C. Similarly, the four CMIP5 simulations are averages from 1860–1890 of the historical runs (using one ensemble member only, r1i1p1) and so the comparisons to the HadCM3 family are not perfectly

clean.

HadCM3-M1 (Fig. 3a) generally has a small cold bias compared to the data, with most regions experiencing colder temperatures of 2 °C to 3 °C. The area weighted root mean square differences (RMSE) is 2.8 °C, but with smaller errors in the tropics and a small warm bias in South America. There is also a small warm bias over Greenland but this should be treated



**Table 3.** Observational and reanalysis datasets used for the evaluation in Fig. 2

| Performance metric | Obs. dataset | Re-analysis dataset | Year(s) for comparison |
|---|---|---|---|
| TOA outgoing All-sky Short wave radiation (rsut_Glob) | CERES-EBAF | — | 2001–2012 |
| TOA outgoing All-sky Long wave radiation (rlut_Glob) | CERES-EBAF | — | 2001–2012 |
| Precipitation (pr_Glob) | GPCP-SG | — | 1979–2005 |
| Near-surface temperature (tas_Glob) | — | ERA-Interim NCEP | 1979–2005 1979–2005 |
| Specific humidity (400-hPa) (hus_Glob-400) | — AIRS | ERA-Interim — | 1979–2005 2003–2010 |
| Geopoential height (500-hPa) (zg_Glob-500) | — | ERA-Interim NCEP | 1979–2005 1979–2005 |
| V-wind height (200-hPa) (va_Glob-200) | — | ERA-Interim NCEP | 1979–2005 1979–2005 |
| V-wind height (850-hPa) (va_Glob-850) | — | ERA-Interim NCEP | 1979–2005 1979–2005 |
| U-wind height (200-hPa) (ua_Glob-200) | — | ERA-Interim NCEP | 1979–2005 1979–2005 |
| U-wind height (850-hPa) (ua_Glob-850) | — | ERA-Interim NCEP | 1979–2005 1979–2005 |
| Temperature (200-hPa) (ta_Glob-200) | — | ERA-Interim NCEP | 1979–2005 1979–2005 |
| Temperature (850-hPa) (ta_Glob-850) | — | ERA-Interim NCEP | 1979–2005 1979–2005 |

with some caution since there are issues about elevation effects and the data is relatively sparse in this region. The results for Fig. 3a, are largely identical to those calculated using the CMIP5 HadCM3-M1 archived data (run by the UK Hadley Centre) for the historical run averaged between 1860–1899 inclusive (not shown). The differences are mostly less than 0.5 °C and





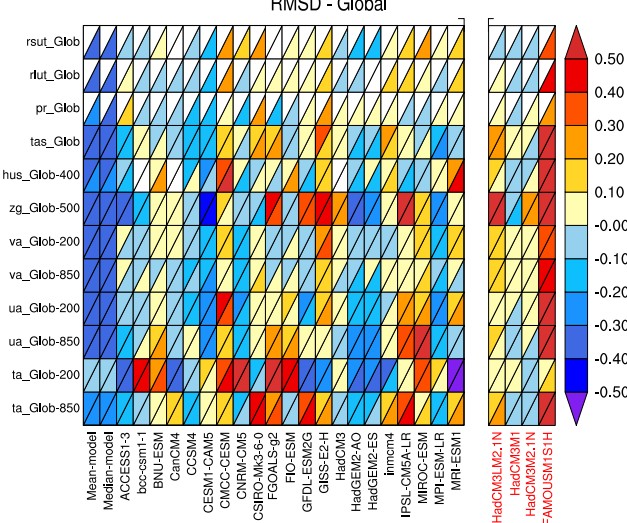

**Figure 2.** Relative error measure of the CMIP5 models (21 in total; in black) and the BRIDGE models (four in total; in red) performance. Error measure is calculated from a time-space root-mean square error (RMSE) of contemporary and predecessor CMIP5 model historical climatological (1980–2005) seasonal cycle simulations and BRIDGE pre-industrial seasonal-cycle climatologies against observations (1980–2005) for a set of nine different atmospheric variables. Error for each individual variable is characterised as a relative error by normalising the result of the median error of all model results (Gleckler et al., 2008), the BRIDGE models are not included in the mean/median error. For instance, a value of 0.20 indicates that a model's RMSE is 20 % larger than the median CMIP5 error for that variable, whereas a value of −0.20 means the error is 20 % smaller than the median error. The diagonal split grid square shows the relative error for the reference observed/reanalysis dataset (lower right triangle) and the alternative dataset (top left triangle). White triangles/boxes indicate where no data was available. Evaluated global atmospheric variables are TOA outgoing All-sky short wave radiation (rsut), TOA outgoing Allsky outgoing Long wave radiation (rlut), Precipitation (pr), near-surface temperature (tas), Specific humidity at 400-hPa (hus_Glob- 400), Geopotential at 500-hPa (zg§_Glob-500), V-wind at 200-hPa (va_Glob-200), V-wind at 850-hPa (va_Glob-850), U-wind at 200-hPa (ua_Glob-200), U-wind at 850-hPa (ua_Glob-850), Temperature at 200-hPa (ta_Glob-200), Temperature at 850-hPa (ta_Glob- 850).

never exceed 1 °C, with an RMSE of 0.5 °C. Differences between the 1860–1889 average and the 1960–1989 average for the CMIP5 historical run are small, verifying that the model biases greatly exceed any differences between pre-industrial and modern temperatures. However, the small warming that does occur between 1860–1889 and 1960–1989 does reduce the cold bias marginally (RMSE decreased by 0.1 °C).

5   Using MOSES 2, HadCM3-M2.1 (Fig. 3b) shows a significant reduction in the cold bias, resulting in a RMSE of 2.1 °C. The cold bias has reduced but still remains over northern Russia and Scandinavia, while over South America (Amazon) and Greenland the warm anomalies have intensified. Over the Amazon this is likely due to the difficulties in the vegetation model (see Sect. 5.3.1), while difficulties with Greenland were mentioned above. Elsewhere, the general cool bias seen in Fig. 3a has gone, replaced by anomalies of ±2 °C to 5 °C, with few widespread regional anomalies. Similarly, HadCM3-M2.2N (Fig. 3c)

10  also shows a reduced cold bias, with an RMSE of 2.1 °C. This model variant shows a slight reduction in the warm anomaly



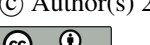

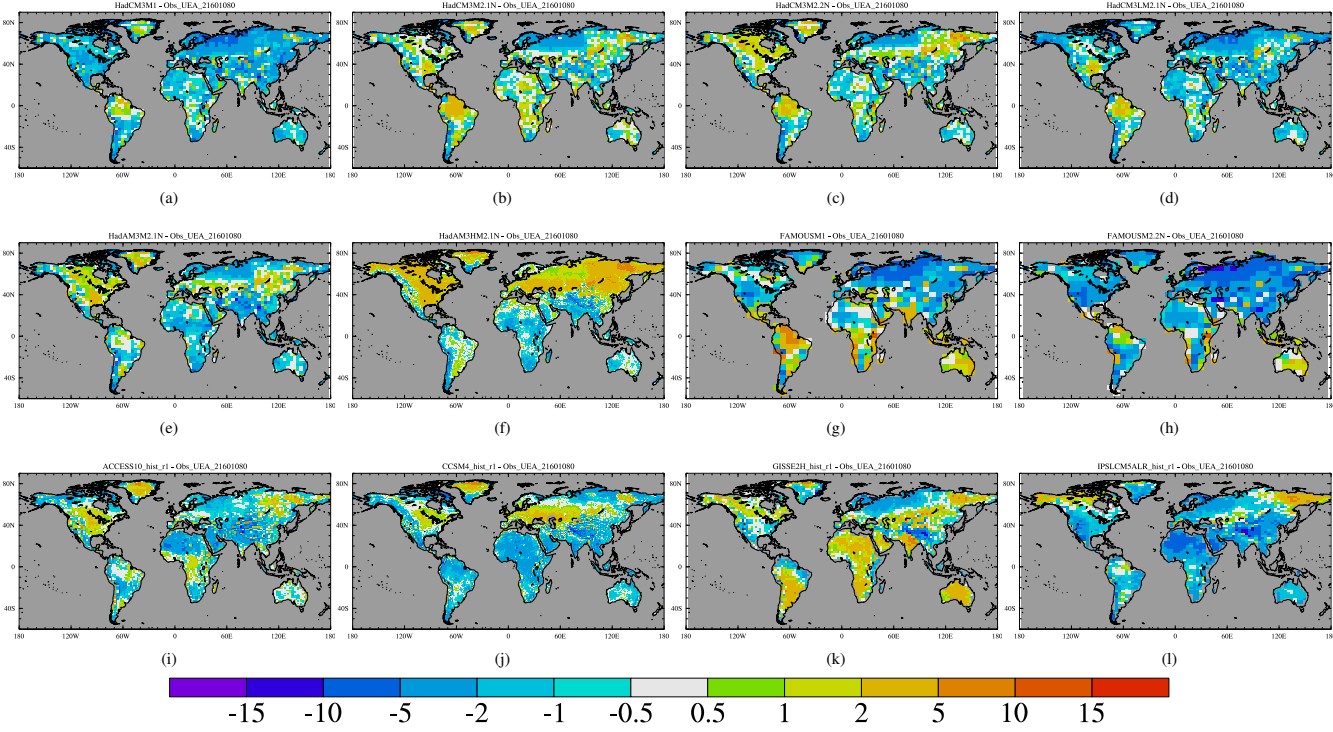

**Figure 3.** (a) The difference between the annual mean surface air temperature (in °C) of our version of HadCM3-M1 and the CRU CL v2.0 for the period 1960–1990 regridded onto the HadCM3-M1 grid, (b) As (a) for the HadCM3-M2.1 version, (c) As (a) but for HadCM3-M2.2, (d) As (a) but for the HadCM3L-M2.1N version, (e) As (a) but for HadAM3-M2.1, (f) As (a) but for HadAM3H-M2.1, (g) As (a) but for FAMOUS-M1, and (h) As (a) but for FAMOUS-M2.2. (i), (j), (k) and (l) show comparable results for four CMIP5 models, ACCESS1-0, CCSM4, GISS-E2-H and IPSL-CM5A-LR respectively. These were chosen to represent two models which were above the CMIP5 average in terms of their RMSE with respect to surface air temperature, and two models which were below average. All differences are calculated by regridding the CRU data onto the corresponding model grid, using simple bi-linear interpolation.

observed over the Amazon compared to Fig. 3b, but has a more extensive warm bias of 1 °C to 2 °C at higher northern latitudes, e.g., over North America.

 HadCM3L-M2.1 (Fig. 3d) has a RMSE of 2.6 °C and a comparable cold bias to HadCM3-M1. As with the HadCM3 model variants, using MOSES2 with HadCM3L reduces the cold bias and RMSE compared to using MOSES1, with HadCM3L-M1 having a much higher RMSE (not shown). Once again, the high northern latitudes (particularly over Russia and Scandinavia) are too cold, which is the result of an exaggerated seasonal cycle due to an overly cold winter. This is also the case for other HadCM3 model variants, but it is most pronounced for the HadCM3L variants. Similarly to the other simulations using MOSES 2, the Amazon remains slightly warmer than the observations with slightly reduced broadleaf forest cover (see Sect. 5.3.1).



The atmosphere-only models vary significantly depending on their resolution. At standard resolution, HadAM3-M2.1 (Fig. 3e) shows similar spatial anomalies and RMSE to Figs. 3a–d, but greater warm biases over North America and Greenland of up to 5 °C and cool biases over Africa and southern Asia of 2 °C to 5 °C. However, it has the smallest anomaly over the Amazon compared to the other standard resolution model variants, and a comparable RMSE (2.3 °C). HadAM3H-M2.1 (Fig. 3f) on the

other hand shows a markedly different spatial pattern in its temperature biases to the model versions already described. It is the only simulation not to show a global cold bias. This is due to warmer than observed temperatures of 2 °C to 5 °C over the majority of land surfaces north of 30° N (with the exception of the southern tip of Greenland and mountainous regions). It has a slight cold bias of 1 °C to 2 °C over areas south of 30° N (with the exception of some regions in South America). Although these biases are extensive spatially, they are not of greater magnitude than the regional biases found in other model variants or

CMIP5 models and the RMSE of the HadAM3H-M2.1 simulation is 2.2 °C.

The FAMOUS model variants (Figs. 3g and 3h) have larger RMSE values than the higher resolution model variants and the other CMIP5 models. FAMOUS-M2.2 (Fig. 3h) is the worse of the two, with a RMSE of 4.1 °C and extreme cold biases over Northern Hemisphere continents, which exceed 10 °C around Scandinavia. The cold bias in Fig. 3g is less extreme, but instead has a warm bias in South America of up to 5 °C to 10 °C and up to 2 °C to 5 °C over India and Australia. There is some

improvement in the RMSE for FAMOUS-M1, but it is still much higher (3.3 °C) than the higher resolution model variants.

For comparison, we show the SAT fields from four CMIP5 models (Figs. 3i–l), selected based on the results of the IPCC AR5 WG1 model evaluation (Chapter 9). We selected two models which were above average for their simulation of SAT (ACCESS1-0 and CCSM4) and two models which were below average (GISS-E2-H and IPSL-CM5A-LR). In all cases these models are not the best or worst extremes, but represent the typical range of model skill. Again, the observations have been

interpolated onto the appropriate resolution of the model from which the RMSE was calculated. As can be seen, the general picture that emerges is that most of the varieties of HadCM3 (except perhaps for FAMOUS) are well within the skill of the CMIP5 ensemble. The CMIP5 models all show large regional biases of up to $\pm 5$ °C (with little consistency on the sign of the anomaly between them) and the RMSE scores range from 2.3 °C to 3.3 °C, which are similar to the varieties of the HadCM3 model. Indeed HadCM3-M2.1N and HadCM3-M2.2 have the smallest RMSE values of the models sampled.

### 5.1.2   Precipitation

Figure 2 shows that the BRIDGE models with the exception of FAMOUS produce annual precipitation amounts comparable to other CMIP5 models suggesting that our models are capturing the general synoptic scale features (frontal, convective and mesoscale).

While global annual RMSE for the BRIDGE models compare favourably it is also key to investigate the mean spatial patterns

of precipitation to ascertain whether the models are reproducing these patterns in accordance with the observations. We assess annual climatological precipitation for the BRIDGE model suite against CRU CL v2.0 (New et al., 2002), a high resolution (0.5°×0.5°) global land surface product (excluding Antarctica). The resolution is transformed (bi-linear interpolation) to the appropriate grid in the model. We are again comparing our pre-industrial simulations with 1960–1990 observations, but the model biases are generally much larger than any trends.




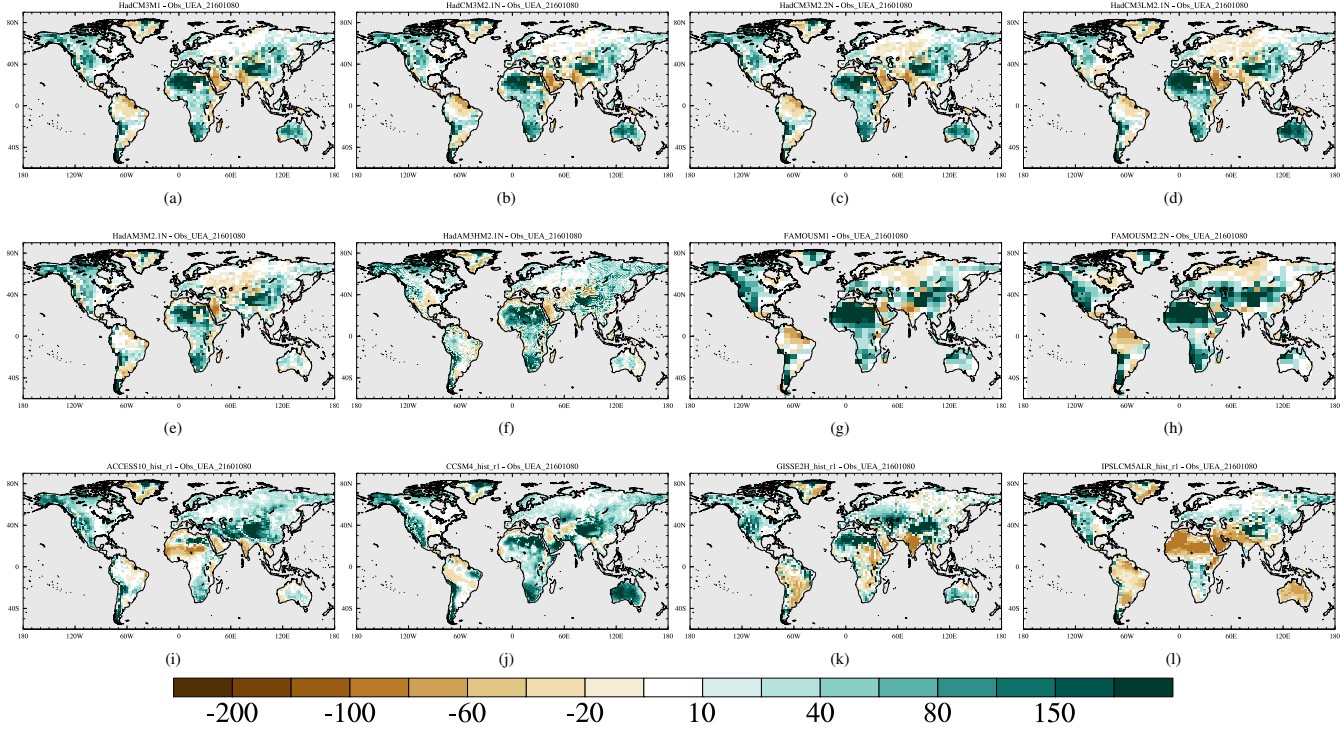

**Figure 4.** As for Fig.3 except showing the difference in mean annual precipitation, expressed as a % difference to the CRU CL v2.0 observations.

Figure 4 shows the regional biases in mean annual precipitation, expressed as a % error compared to the CRU CL data. For consistency with the previous figure, we also include the same four CMIP5 models. Regionally, spatial patterns in precipitation bias are generally consistent between the different BRIDGE models and broadly comparable to their CMIP5 models.

The BRIDGE simulations affirm a known problem in all CMIP5 models whereby they overestimate precipitation in regions

5  of steep topography. This may be due to poor representation of moisture gradients and regional dynamics, but can be amplified by a known negative bias in gauge stations amplifying model-observation disparity (Adam et al., 2006). Secondly, they also underestimate precipitation leeward of the mountain range (e.g., Himalaya's and Andes), as well as over arid regions, again a known bias in the CMIP5 model suite.

Monsoonal regions of south-east Asia, Austral-Asia, southern South America, West and central Africa, overestimate precipi-

10  tation by $0.5\,\mathrm{mm\,day^{-1}}$ to $2\,\mathrm{mm\,day^{-1}}$ while underestimate precipitation in the Indian and northern arm of the South American region by $\sim 1\,\mathrm{mm\,day^{-1}}$ to $4\,\mathrm{mm\,day^{-1}}$. There are however some exceptions with HadAM3-M2.1 (Fig. 4), an atmosphere-only GCM using observed SST (HadISST 1870–1900) producing a more reasonable precipitation signal compared to the observations suggesting the importance in the accuracy of SST/local ocean circulation dependency (this is also seen in Australia).



There is still a problem with the ITCZ location over South-America being too far South giving this north-south dipole in negative/positive anomalies.

It is also noted that an increase in resolution does not produce a noticeable improvement on spatial annual precipitation bias in certain monsoon regions in the BRIDGE models (Fig. 4f compared to 4j) again suggesting the importance of accuracy in SST and ocean circulation, with the exception of South America where there is improvement. Spatially, increased resolution does not effect the sign of anomaly nor the spatial patterns of precipitation regionally (with the exception of South America) throughout the BRIDGE suite of models however the magnitude of the precipitation bias does progressively decrease.

### 5.1.3 Top of the atmosphere radiation fluxes

There is broad agreement between the observed and simulated total northward heat transport. Similarly, the partitioning between the ocean and atmosphere is qualitatively similar to that estimated by Trenberth and Fasullo (2013). We find that all versions of the model simulate heat transport that are consistent with CMIP5 models (see grey lines in Fig. 5). However, in common with almost all other climate models we find that on the equator, although the total heat transport is northward, in agreement with the observations, the atmospheric heat transport is also northward, contrary to the observed southward transport (Loeb et al., 2016). The cause of this in any of the models in which it is a feature is unclear. The three versions of HadCM3 show remarkably similar amounts of total heat transport, the major difference is FAMOUS which underestimates the southward heat transport in the Southern Hemisphere subtropics rather more than HadCM3 and HadCM3L. This is due to the smaller amount of ocean heat transport in this region in FAMOUS. This discrepancy is not due to the coarse resolution of the FAMOUS ocean because, interestingly, in this region the ocean heat transport in HadCM3L is very similar to HadCM3 whose ocean resolution is quite different. Therefore it is more likely that the difference arises from the atmospheric forcing of the surface ocean. In the Northern Hemisphere the HadCM3L ocean heat transport is more similar to AMOUS, suggesting that the ocean resolution is more important here. This is likely due to the processes that determine the ocean's overturning circulation being simulated rather differently in the higher and lower resolution models.

## 5.2 Ocean

### 5.2.1 Sea Surface Temperature

The BRIDGE suite of models is capable of reproducing the broad global latitudinal patterns and gradients in SST (Fig. 6). Nonetheless, some cold and warm biases of over 8 °C are present especially where sharp fronts and boundary currents are not resolved. Other biases of similar magnitude also appear in the upwelling regions (e.g., west of Africa and of South America), and again these are likely associated with processes that are not fully resolved by the model. Colder SSTs in the sub-polar North Atlantic for all models are not uncommon and likely due to the coarse resolution (e.g., Marzocchi et al., 2015a). This can be seen by comparing HadCM3 and HadCM3L, which are models that differ most in their ocean resolution. Cold biases in the Northern Hemisphere are more extensive in HadCM3L than in HadCM3. Warmer SSTs of up to 8 °C are present in the Southern Hemisphere, especially in the Southern Ocean, both in HadCM3 and HadCM3L. FAMOUS is characterised by




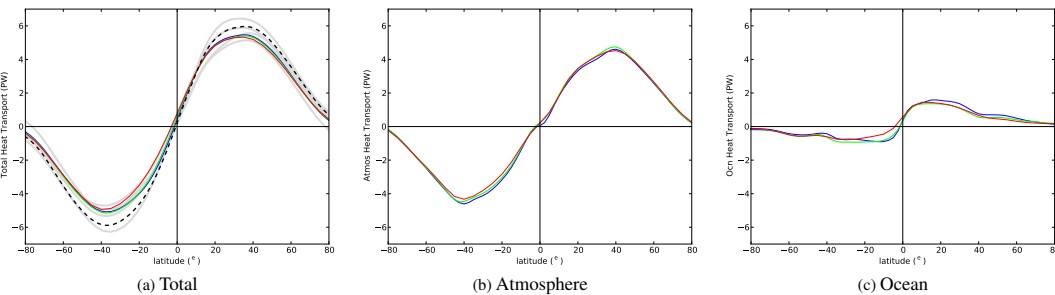

| (a) Total | (b) Atmosphere | (c) Ocean |

**Figure 5.** The annual mean northward heat transport in total, the atmosphere and the ocean. Black line shows the observational estimate, blue: HadCM3-M2.1, green: HadCM3L-M2.1 and red: FAMOUS-M2.2. These transports are calculated as the implied heat transports from the TOA and surface energy fluxes (see Trenberth and Fasullo (2013) for details). Observational estimates for the total transport are derived from the CERES data.

colder than observed SSTs in the Northern Hemisphere, in common with HadCM3 and HadCM3L, and warmer SSTs by up to 8 °C almost everywhere in the Southern Hemisphere. HadCM3 and HadCM3L do not show any notably larger biases when compared to typical CMIP5 models. All of the HadCM3 models, including FAMOUS, show smaller temperature biases in the Southern Ocean than GISS-E2-H, and the biases in the North Pacific are of a similar magnitude to those in IPSL-CM5A-LR.

### 5.2.2 Sea Surface Salinity

The broad global latitudinal patterns of sea surface salinity are realistically reproduced by the suite of BRIDGE simulations (Fig. 7). However in the global average, the models show a fresh bias of about 0.5 g kg$^{-1}$, as we shall show in the following section this is likely related to the rather different vertical structure of the ocean in the model than in the observations. In all models, substantial differences from the observations are found in the Arctic Ocean, exhibiting higher salinities (up to 10 g kg$^{-1}$) in the Kara Sea and generally north of Russia. Generally lower salinities (of up to 5 g kg$^{-1}$) are found in the Chukchi and Beaufort seas. The largest differences are found in enclosed or semi-enclosed basins, such as the Mediterranean Sea, where it is more saline, or the Black Sea, Caspian Sea and Hudson Bay where it is markedly fresher. In all versions of the model the subtropical North Atlantic tends to be more saline than the observations.

Substantial differences from the observations can also be found in CMIP5 models (Figs. 7g–h) with magnitudes comparable to the BRIDGE models. We note that some of the differences at high latitudes could be due to biases in the simulation of sea ice concentration and distribution.

### 5.2.3 The Atlantic Meridional Overturning Circulation

Figure 8 shows the mean strength of the Atlantic Meridional Overturning Circulation (AMOC) for the three main model families (HadCM3, HadCM3L and FAMOUS). Values are shown as zonally integrated depth profiles measured in terms of the northward flow of water at 26.25° N. The modelled AMOC is compared to observations from the Rapid Climate Change-





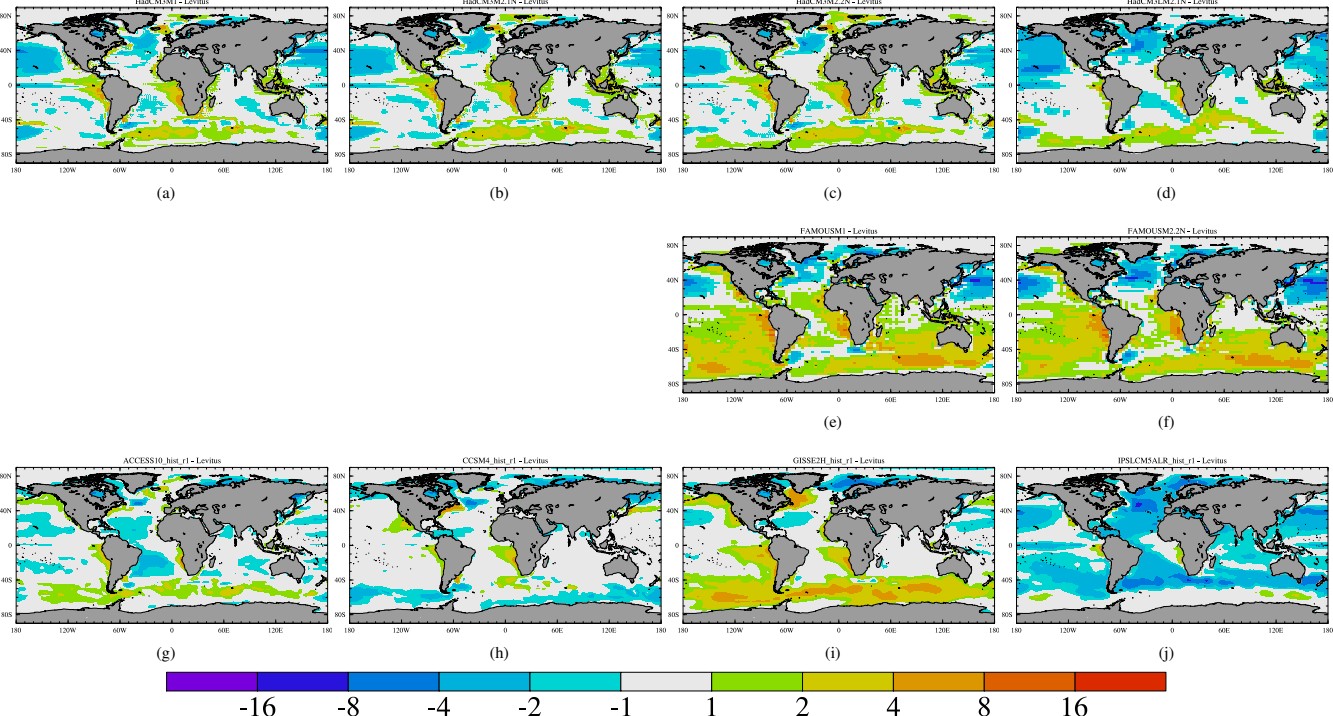

**Figure 6.** Annual mean sea surface temperature differences (in °C) for a range of coupled model simulations, and also for the same four CMIP5 models used in Figs. 3 and 4. The observational dataset is the Levitus World Ocean Atlas (2009) (Locarnini et al., 2010). The figure shows the difference in SST between model and observations for (a) HadCM3-M1, (b) HadCM3-M2.1, (c) HadCM3-M2.2, (d) HadCM3L-M2.1, (e) FAMOUS-M1, (f) FAMOUS-M2.2, (g) ACCESS1.0, (h) CCSM4, (i) GISS-E2-H, and (j) IPSL-CM5A-LR. Model output is regridded to the same resolution of the observations.

Meridional Overturning Circulation and Heatflux Array (RAPID-MOCHA) at 26.5° N (Smeed et al., 2015), which have been calculated from daily data spanning the 2nd April 2004 to the 30th March 2015.

The strength of the AMOC varies on an annual basis so a range of values is shown for both the models and observations, with the depth at which the AMOC peaks highlighted with a point. The peak flow of the North Atlantic Deep Water (NADW) cell identified by the RAPID-MOCHA array lies at around 1000 m and varies annually between 14 Sv and 19 Sv. All three models do a reasonable job at modelling the NADW cell in terms of the magnitude of maximum flow. However, maximum overturning is too shallow for all model variants, peaking at approximately 800 m. HadCM3L has a larger annual variation, approximately twice as large as that of the observations, resulting in an underestimation of the annual minimum peak northward transfer. FAMOUS model variants tend to underestimate the annual variation by approximately 50 %, while HadCM3 variants have a realistic annual variation at least in the upper 1500 m of the ocean.





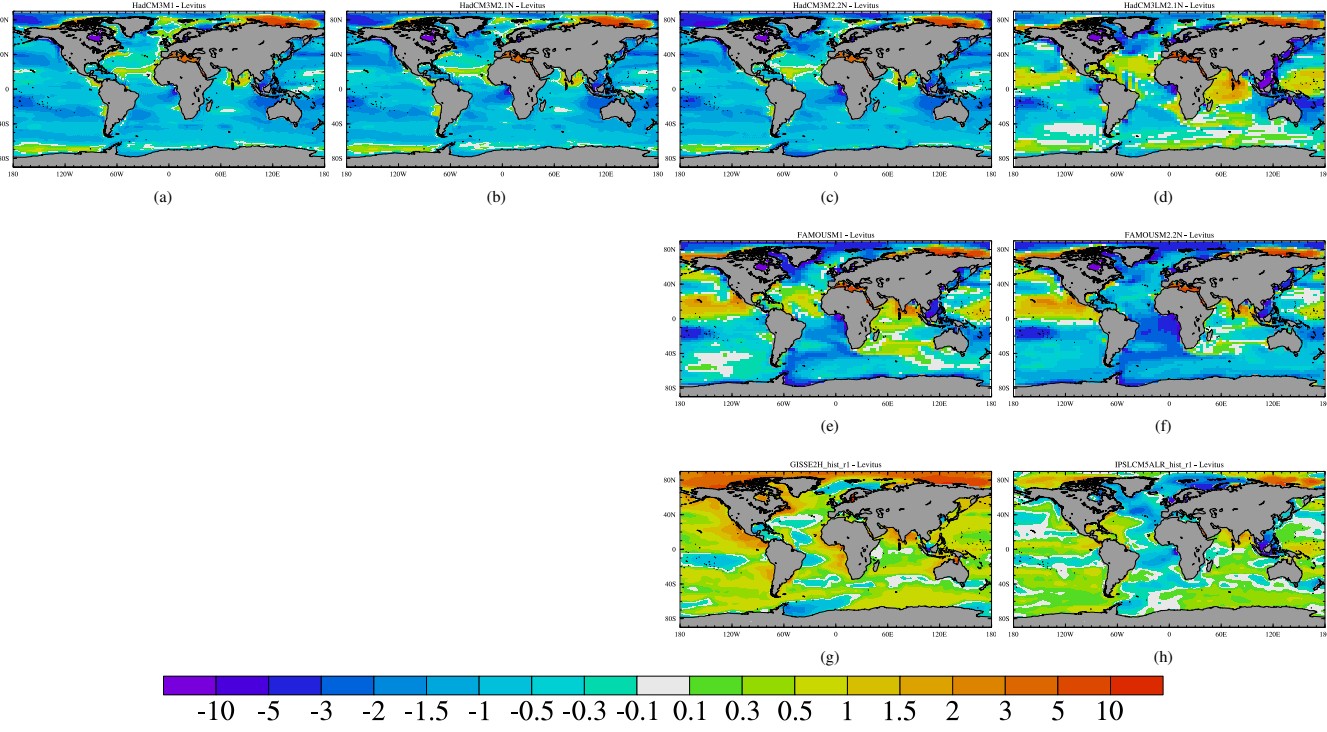

**Figure 7.** As Fig. 6 but showing the differences in sea surface salinity (in $g\,kg^{-1}$) between models and observations. (a) HadCM3-M1, (b) HadCM3-M2.1, (c) HadCM3-M2.2, (d) HadCM3L-M2.1, (e) FAMOUS-M1, (f) FAMOUS-M2.2, (g) GISS-E2-H, and (h) IPSL-CM5A-LR. The observational dataset is the Levitus World Ocean Atlas (2009) (Antonov et al., 2010). Model output is regridded to the same resolution of the observations.

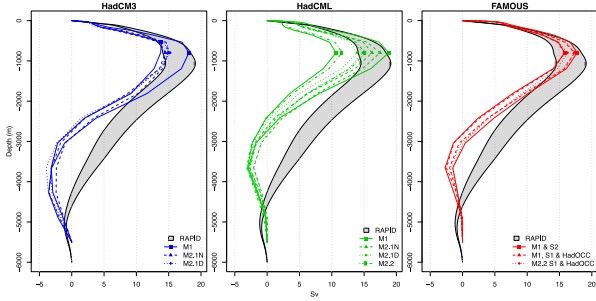

**Figure 8.** Annual depth profiles of the Atlantic meridional overturning circulation (AMOC) at 26.25° N showing range of values for variants of HadCM3, HadCM3L and FAMOUS. Annual data from the RAPID array at 26.5° N is highlighted in grey. The depth at which the AMOC reaches its maximum is indicated with a point.



All of the models do a poor job at representing the flow of the NADW cell below 2000 m depth. McCarthy et al. (2012) showed that at this latitude, approximately 60 % of the southward return flow is comprised of upper NADW (between 1100 m and 3000 m) and 40 % of the lower NADW (between 3000 m and 5000 m). The modelled stream functions show that the return flow is shifted to shallower depths, indicating a shallower overturning in all of the model variants.

The CMIP5 models exhibit a wide spread in the mean strength of the AMOC, ranging from 13 Sv to 31 Sv and peaking at latitudes between 20° N and 60° N (e.g., Zhang and Wang, 2013). It was not possible to include the CMIP5 models in Fig. 8, however, the studies of Roberts et al. (2013) and Msadek et al. (2013) produced similar plots of AMOC zonally integrated depth profiles for a range of models compared to observations (their Figures 1 and 3 respectively). The HadCM3 and FAMOUS variants are shown to have very similar stream function profiles to GFDL Climate Model 2.1, NCAR CCSM4 models and the

MPI models, and more accurately simulate the maximum overturning than the NorESM1 model variants. A similar pattern of biases is apparent in the vertical structure for these models, i.e., a too shallow overturning cell, however the point of maximum overturning is shallower in the HadCM3 and FAMOUS variants.

This bias in the vertical structure has been attributed in some studies to inaccurate transport in the Nordic Sea overflows, which in the case of HadCM3 includes a greater than observed overflow across the Denmark Strait, in addition to subgrid scale

processes (see Legg et al., 2009; Roberts et al., 2013).

An additional cause of the shallow overturning may be the excessive surface salinity in the North Atlantic in all model versions, particularly around the subtropics as shown in Fig. 7. The study of Pardaens et al. (2003) investigated the freshwater budget in HadCM3, concluding that in the North Atlantic saline conditions are primarily a result of excessive evaporation. Other components, such as insufficient subtropical runoff from the west coast, may also have an influence. This results in the

Atlantic being too stratified and consequently too stable which may reduce the depth of overturning.

A further consequence of this is a net northward transport of fresh water into the Atlantic (Liu et al., 2014), which is thought to result in a monostable stability regime of the AMOC in HadCM3 instead of a bistable regime (Weaver et al., 2012; Liu et al., 2014). Approximately 60 % of the CMIP5 models have been shown to exhibit monostability (Weaver et al., 2012). However this is contrary to what is indicated in observations; that there is a net export of freshwater from the Atlantic and consequently

the AMOC is in a bistable regime. This indicates that the AMOC may be artificially stable in the HadCM3 and FAMOUS model variants in addition to a range of other CMIP5 models.

### 5.3  Land

### 5.3.1  Vegetation Distribution

These models have a simple representation of terrestrial vegetation, with five plant functional types that each covers a large

climatic range. Comparing the dominant PFT in the model to a reconstruction of pre-industrial vegetation (Ramankutty and Foley, 1999), we can see the model captures the overall correct pattern (Fig. 9), with slight errors of extent and/or exact location. Previous studies (Betts et al., 2004) which compared TRIFFID PFT distributions to the IGBP-DIS land cover dataset (which





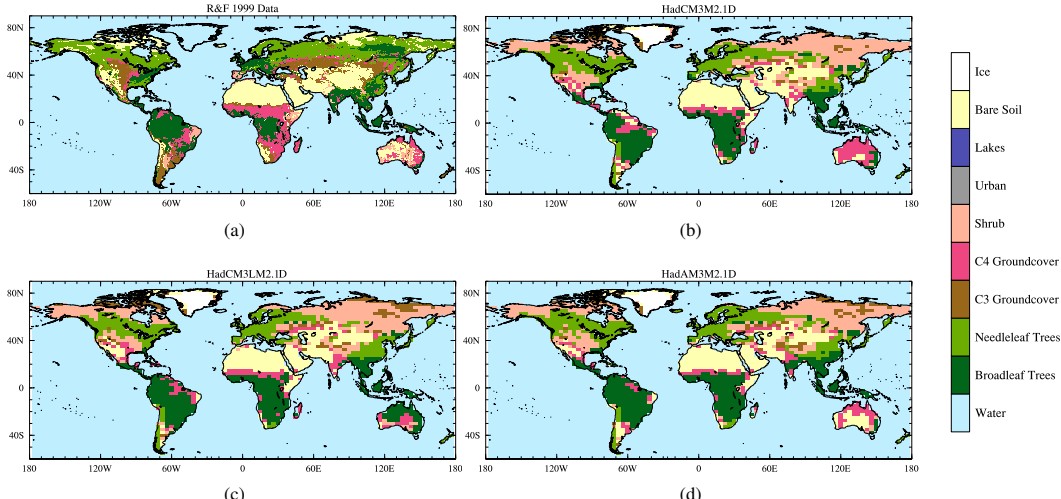

**Figure 9.** Maps of the dominant plant functional type for observations (a) and model simulations of the pre-industrial. The models shown are (b) HadCM3-M2.1D (c) HadCM3L-M2.1D and (d) HadAM3-M2.1D. The observed dataset for comparison is Ramankutty and Foley (1999).

represents the modern distribution of vegetation as derived from satellite image interpretation, Loveland and Belward (1997) found much of the same patterns.

The broadleaf trees in the models have a known problem of extending too far, especially in the Southern Hemisphere, and this can be seen in Fig. 9. The southern mid-latitudes are difficult to capture accurately, for a variety of reasons, including the challenge of precipitation patterns in this region. The HadCM3L model is significantly worse than either HadCM3 or HadAM3 in this regard, because of its decreased ocean resolution.

A feature which is isolated to this version of HadCM3 is a tendency for the Amazon broadleaf forest extent to be underestimated at the mouth of the River Amazon, even at relatively low carbon dioxide concentrations. At higher carbon dioxide levels, this is a known feature of the model (Malhi et al., 2009; Betts et al., 2004) caused by ocean circulation resulting in insufficient precipitation to sustain the forest. Since the model underestimates precipitation, this leads to TRIFFID modelling the presence of C4 grasses instead of broadleaf trees.

Grasses tend to be globally slightly underestimated with the position of vegetation in the Sahara desert and other arid regions well reproduced, but the density is modelled to be too sparse, particularly in south-west Africa, central and south-west Asia, south-west North America and Australia.

The shrub PFT is overestimated at high latitudes, perhaps as a result of the high latitude cold bias in the model. We can see this in these simulations (Fig. 9b). There is the correct pattern of needleleaf trees, but lower amounts than observations for Ramankutty and Foley (1999). However, in Fig. 9e, we can see that the models consistently overestimate the amount of high northern latitude shrubs, mainly because of underestimation of bare ground (Fig. 9f).



The observational dataset is re-gridded from the original to the nine surface types in our models, which introduces more uncertainty. In particular, the dominant PFT obviously is a difficult metric to consider precisely, as it does not represent mixed vegetation systems such as Savannah, well. Some difficulties mainly originate in how areas such as tundra are allocated – to bare soil or to C3 grasses. Because of the limited number of PFTs in the model, C3 grass represents a large range of low-lying
vegetation types, arguably also encompassing mosses and lichen and very sparse tundra vegetation.

There are also some uncertainties associated with the Ramankutty and Foley (1999) dataset, which is a reconstruction of pre-industrial vegetation. Other model-observations discrepancies have been suggested to be a combination of orographic representation leading to underestimation of precipitation and the inadequate treatment of natural disturbance mechanisms such as fire (Betts et al., 2004).

Though not shown, the Equilibrium simulations are very similar to the Dynamic ones, especially in the tree PFTs. That the Equilibrium and Dynamic simulations from the same model are very closely related suggests that although the inter-annual variability does have some influence on the vegetation, in general the mean climate is more important.

### 5.3.2   Net Primary Productivity

The Net Primary Productivity (NPP) of the models, compared to MODIS 2001 NPP observations, is good at capturing the
global latitudinal patterns, with higher NPP in the tropics and lower in other regions (see Fig. 10). One notable exception is the failure of the model to capture sub-tropical spikes in productivity and the peak for the Amazon tropics area is lower in the model than observations. However, the NPP performance of our models compares favourably with that of CMIP5 models. The large range of NPP values of these CMIP5 models encompasses our models at nearly all latitudes.

The Amazon forest extends a little too far south in all the models, but this is a key area of difference as well, with HadAM3
models better capturing the observed distribution, and the lower resolution ocean of HadCM3L suffering the most from excess tropical forest. However, HadCM3L models do better in the Southern Hemisphere, and better than HadCM3 in other parts of the tropics. As in the case of the PFT distribution (upon which the NPP is based) there is a close relationship between the equilibrium (not shown) and dynamic simulations of NPP.

## 6   Summary and Future Directions

This paper provides an overview of a variety of versions of the HadCM3 family of coupled climate models used in BRIDGE at the University of Bristol. We provide updated documentation of these variants, including atmosphere-only, low-resolution ocean, and high-resolution atmosphere-only models, and including three alternative versions of the MOSES land surface scheme. Using an up-to-date set of observational benchmarks we show through detailed comparisons, that the models provide a good representation of large-scale features of the climate system, both over land and for the ocean and which remains
comparable to most CMIP5 models.

The speed and relative complexity of HadCM3 and its variants creates opportunities for tackling a range of problems. Large ensembles are possible because of the relatively small number of processors required. Ensembles can explore probabilistic ap-





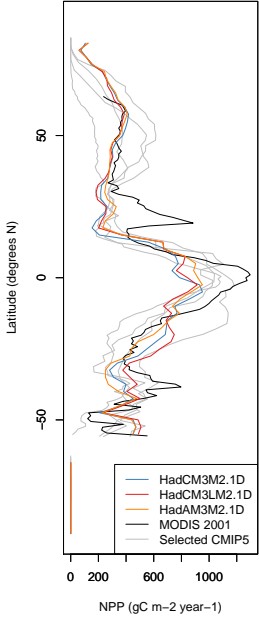

**Figure 10.** The latitudinal average NPP in $g_C$ m$^{-2}$ year$^{-1}$. CMIP5 models without dynamic vegetation plotted here are: CCSM4 and IPSL-CM5A-LR. CMIP5 models with dynamic vegetation plotted here are: MIROC-ESM and MPI-ESM.

proaches to climate change quantification, model parametric uncertainty or boundary condition uncertainty. Long integrations of many millennia are also possible, so that longer term climate changes, for example covering the last deglaciation, can be investigated.

Several versions of the model are under continued development and improvement. For example, FAMOUS has been coupled
to an interactive ice-sheet model (Gregory et al., 2012) to allow predictions of sea-level and land ice on longer timescales. Further developments in this approach will allow more detailed investigation of the climate – sea-level interactions for a variety of times in the past (e.g., Roberts et al., 2014). FAMOUS also now includes a marine carbon cycle (HadOCC) (Williams et al., 2013) and an oceanic oxygen cycle (Williams et al., 2014), allowing direct comparisons to biogeochemical cycles.

Currently a very high-resolution version of HadAM3H is finalising development in Bristol. This uses a resolution of
$0.625° \times 0.4166°$ ($576 \times 433$ grid points, N288) as this has been suggested as a minimum resolution for realistic simulation of the hydrological cycle (Demory et al., 2014). The model appears to be significantly computationally more efficient (approximately $10\times$ faster) than a similar resolution version of the more recent UK Met Office HadGEM3 model (Walters et al., 2014), because of the lower model top, simplified aerosol physics and major differences in the underlying atmospheric dynamical core.

This paper motivates the continued development and scientific application of the HadCM3 family of coupled climate models.
Future updates will cover new developments to the presented model version, bug corrections and enhancements.



## 7 Code Availability

The UK Meteorological Office made available the source code of HadCM3 via the Ported Unified Model release (http://www.metoffice.gov.uk/research/collaboration/um-partnership)

The main repository for the Met Office Unified Model (UM) at the version corresponding to the model presented here can be found at http://cms.ncas.ac.uk/code_browsers/UM4.5/UMbrowser/index.html.

The code detailing the advances described in this paper is completely contained within the files available as a Supplement to this paper. These files are known as code modification files or "mod" files and should be applied to the original code of the model. This is protected under Crown Copyright, as is the base code linked above.

The UM basis files for the simulations described in this paper can be found on the puma.nerc.ac.uk facility (please contact Andy Heaps for access, andy.heaps@ncas.ac.uk). The simulation names are:

- tcsyf: HadCM3-M1

- tcywd: HadCM3-M2.1N

- tcyxc: HadCM3-M2.2N

- tdbad: HadCM3L-M2.1N

- tdekd: HadAM3-M2.1N

- tdewb: HadAM3H-M2.1N

- tdexb: FAMOUS-M1

- tdeyb: FAMOUS-M2.2N

- tdkym: HadCM3-M2.1D

- tdkyn: HadCM3L-M2.1D

- tdkyo: HadAM3-M2.1D

## 8 Data Availability

- The CERES data were obtained from the NASA Langley Research Center CERES ordering tool at (http://ceres.larc.nasa.gov/).

- We acknowledge the World Climate Research Programme's Working Group on Coupled Modelling, which is responsible for CMIP, and we thank the climate modelling groups for producing and making available their model output. For CMIP the U.S. Department of Energy's Program for Climate Model Diagnosis and Intercomparison provides coordinating



support and led development of software infrastructure in partnership with the Global Organization for Earth System Science Portals.

- We acknowledge the MODIS/TERRA project for the NPP data, acquired from: http://neo.sci.gsfc.nasa.gov/view.php?datasetId=MOD17A2_M_PSN&year=2001.

- Data from the RAPID-WATCH MOC monitoring project are funded by the Natural Environment Research Council and are freely available from: http://www.rapid.ac.uk/rapidmoc

*Acknowledgements.*

- Climate simulations were carried out using the computational facilities of the Advanced Computing Research Centre, University of Bristol (http://www.bris.ac.uk/acrc) (Bluecrystal).

- E. Armstrong was funded by NERC (NE/L501554/1).

- M. P. S. Badger was funded by NERC (NE/J008591/1).

- T. Davies-Barnard was funded by the European Commission's Seventh Framework Program grant agreement 282672 (EMBRACE) and EU grant ERC-2013-CoG-617313 (PaleoGenie).

- P. O. Hopcroft was funded by NERC (NE/I010912/1).

- A. T. Kennedy was funded by NERC (NE/L002434/1).

- N. S. Lord was funded by RWM Limited via a framework contract with Amec Foster Wheeler, who were supported by Quintessa.

- A. Marzocchi was funded by the National Science Foundation (NSF) award #1536454

- W. H. G. Roberts was funded by the Leverhulme Trust.

- G. J. L. Tourte was funded through the advanced ERC grant 'The Greenhouse Earth System" (T-GRES, project reference 340923),
awarded to R. D. Pancost.

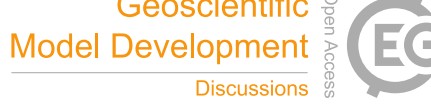

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
