# Peer review of "The BRIDGE HadCM3 family of climate models: HadCM3@Bristol v1.0"

_Geoscientific Model Development, 2017_

## Referee Comment (RC1) · Anonymous Referee #1 · 14 Mar 2017

The purpose of this paper is to document the GCMs derived from HadCM3 that have been and will continue to be used by the authors for a wide range of climate research activities. This is an appropriate purpose for a GMD paper. The paper is clearly and well-written and serves the purpose. Its comparison of the results from various HadCM3-related configurations with each other and with CMIP5 models, to give an overview, is particularly useful. Although HadCM3 is quite an old model, it is still important. In the Web of Science, the paper of Gordon et al. (2000) on HadCM3 has been cited at least 100 times per year in the last decade.

I have first some general comments, then some specific ones. I'm sure that my comments can be addressed, if the editor agrees that they should be, but since they're quite extensive I'm classifying them as implying major revisions.

[Figure]

General comments

(1) It's rather unusual for a paper to document a model that was mostly not developed by the authors, especially when other papers already exist about the model. Moreover the authors are not the sole users of this model. Of course the authors are aware of these points, and have cited papers other than theirs; I have suggested some more citations, and some minor rephrasing to avoid misinterpretation. In a way, this situation is like an open-source development (these models aren't open-source, although readily available). In view of this, the paper should focus on the aspects which BRIDGE specifically has contributed. For example, while the summaries of sects 2.1-2.4 of HadCM3 are clearly written, I feel that this much information is not needed, because it's essentially documentation of the original HadCM3, for which Pope et al., Gordon et al., and references therein should be cited. For the purpose of this paper, sect 2.5 is the important part, giving the differences of the BRIDGE HadCM3-M1 from the model of Gordon et al. These are stated to have very little effect.

(2) The nomenclature of the models could be confusing. The purpose of this paper is to describe models used at BRIDGE, but these have to be clearly related to or distinguished from other models, in the same family, that have been used at the Met Office and elsewhere.

HadCM3 and HadAM3 are models developed by the Hadley Centre. On p3 line 12-14, the authors write, "Since its introduction, HadCM3 (and related models) has undergone a substantial number of changes, bug fixes and adaptations, such that few of the versions of the model used now are truly identical to their original model description in Pope et al. (2000) and Gordon et al. (2000)." I think this statement is debatable. It is likely that Hadley Centre users of HadCM3, and at least some other users, would say that it is not *really* HadCM3 if it's different from the model described by the papers of 2000, as far as the AOGCM is concerned, except for bug-fixes and mods needed for porting. That is, HadCM3 has not been revised. Of course it's hard to decide what's a bug-fix versus a scientific change - that's why this is debatable.

It turns out that the BRIDGE HadCM3-M1 is practically identical to the model of Gordon et al. (2000) (section 2.5). This is useful to know, but I'd suggest that it means it would be clearer to call this model just "HadCM3" throughout the paper. However, we learn that the BRIDGE HadCM3L is a very different model from the earlier Met Office model HadCM3L (p14 line 32). (A reference should be given for the Met Office HadCM3L - I think it might have been first used though not named by Cox et al., Nature, 2000). Similarly, the BRIDGE HadAM3H is said to be different from HadAM3H as used by the Met Office (p14 line 24-25). It is therefore confusing to call the Met Office and BRIDGE models by the same names. At least in the context of this paper, I suggest that models from BRIDGE and the Met Office are given different names, so that the reader can be clear, when differences from HadCM3 are mentioned, which are the ones introduced at BRIDGE, and which are the same as in the Met Office version.

For HadRM3, is Hudson and Jones (a technical report on a particular application) the only reference that can be cited? Perhaps one of the early journal articles using this model describes it? How does the BRIDGE HadRM3 relate to HadRM3P of Jones et al. (2004) [Jones, R. G., Noguer, M., Hassell, D. C., Hudson, D., Wilson, S. S., Jenkins, G. J., and Mitchell, J. F. B.: Generating high resolution climate change scenarios using PRECIS, Met Office Hadley Centre, Exeter, UK, 40 pp., 2004] and to the HadRM3H and HadAM3H of Arnell et al. (JGR, 2003).

There are several published versions of FAMOUS. It would be useful if the authors could relate the BRIDGE version to those documented by Jones et al. (2005), Smith et al. (2008) and Smith (10.5194/gmd-5-269-2012, 2012). The Smith papers identify versions of FAMOUS by the run ID of the definitive UM basis file, so it would be useful to relate the runs detailed in Sect 7 to theirs. Also, FAMOUS is documented at www.famous.ac.uk, which could be cited.

(3) Effect of differences in configuration. It seems to me that there's not a very close connection between the model documentation of sect 2-4 and the results of sect 5. It would be valuable, wherever possible, to relate the differences in results to the dif-

ferences in formulation. I appreciate that some of this is done, and that it's hard, but whatever more can be done would be useful, since it's the kind of information that would be relevant to deciding or understanding the choice of model for a particular purpose.

(4) Computational cost (i.e. speed) is an important consideration in the choice of model. If one had infinite computing resource, for example, one would probably not prefer FAMOUS to HadCM3. It would be informative to collect some numbers for the relative computational cost of the model e.g. in Table 1.

Specific comments

p1 line 8. I would delete "originally". Subsequent developments are smaller changes than the original development; HadCM3 is still essentially the same.

p1 line 9-10. "but is now largely being replaced by more recent models" strikes me as an odd thing to say. It's true that HadCM3 is now little used by the Hadley Centre, which mostly uses later Met Office models. Some other centres, including BRIDGE, use HadCM3, but these centres, including BRIDGE, use other and later models too. Most centres, including the Hadley Centre, routinely use a range of models, older and newer, for different purposes. For reasons the paper explains, there are purposes for which older models are more suitable and cannot be replaced by later ones. I would suggest deleting this phrase.

p1 line 10. This paper is about the models used at BRIDGE, but other places use HadCM3 in the Met Office version. This sentence and the next two could seem to imply that BRIDGE is responsible for development of these models. To avoid that implication, you could slightly rephrase e.g.

It continues to be used by various institutions, including by the BRIDGE (Bristol Research Initiative for the Dynamic Global Environment) research group at the University of Bristol, who have made adaptations over time to the base HadCM3 model. These adaptations mean that the original documentation is not entirely representative of the

models used at BRIDGE, where several other configurations are in use which now differ from the originally described model versions.

p1 line 17. In the title and here you mention "version 1.0" for this suite of models, but you don't mention it again in the paper, which describes each of the configurations individually. Is it useful to assign a version number to the entire suite? If it is, it would be worth explaining what defines a version of the suite, and what would qualify as a new version.

p2 line 4. It would be worth spelling out that "model" is a general circulation model. Not all climate models are GCMs.

p2 line 6. The statement is correct that the HadCM3 family is in use at BRIDGE, but it could seem to imply that BRIDGE is the only group which uses HadCM3 etc. still. As the authors know, and have stated at p3 line 14, it has other active users too.

p2 line 7. While it's true that Gordon et al. (2000) describe developments to the ocean model, I would say that main achievement and interest of that paper is HadCM3 i.e. the coupled model. Thus I would suggest phrasing this slightly differently e.g. "HadAM3 (Pope et al., 2000). Together with improvements to the ocean GCM, this enabled the development of HadCM3 (Gordon et al., 2000), which was one of the first coupled atmosphere-ocean GCMs that did not require ...".

p2 line 9. Replace "was" with "has been", since it's still in use. It has been used for the past as well as the future. I think that at this point it would be appropriate to cite some of the major uses of HadCM3 that don't come up later, such as detection and attribution (Stott et al., Science, 2000), unforced variability (Collins et al., Clim Dyn, 2001), climate projection (Johns et al., Clim Dyn, 2003), uncertainty and constraints on projections (Stott and Kettleborough, Nature, 2002), decadal prediction (Smith et al., Science, 2007). I've made other suggestions later too.

p2 line 10. Why is "though" at the end of the sentence?

p2 line 16. Reichler and Kim (BAMS, 2008) could be cited here as well.

p2 line 17. UKESM hasn't been introduced.

p2 line 24. Hewitt et al. (GRL, 2001) could be cited here as the first use of HadCM3 for a snapshot of palaeoclimate.

p2 line 26. Faster models are also useful for investigation of anthropogenic change on long timescales; HadCM3 has been used for that as well e.g. Gregory et al. (GRL, 2004), Ridley et al. (Clim Dyn, 2005).

p2 line 25. Although multi-centennial rather than multi-millennial, I think it would be relevant to cite Tett et al. (Clim Dyn, 2007) as the first use of HadCM3 for a study of long-term past climate change. Among examples of more recent uses of HadCM3 for such a purpose, I would say that Schurer et al. (Nature Geosci, 2013) could be cited.

p2 line 27-29. What boundary conditions are meant here? I usually understand BCs as concerning forced climate change. Of course there are numerous earlier examples of climate-change experiments with HadCM3. Large initial-condition ensembles are a good application too, I agree, but I would say that a greater use of HadCM3 has been for parameter perturbation ensembles, the next point in this para, so maybe the order of these should be reversed.

p2 line 29-31. I'm unclear what distinction is being drawn between "multiple runs to explore the sensitivity" and "calculating probability density functions". Could these be merged? Murphy et al. (Nature, 2004, the first QUMP paper), and Stainforth et al. (Nature, 2005, the climateprediction.net paper) should be cited.

p3 line 9. Jones et al. (Clim Dyn, 2005) should be cited here as well, since it was the first publication of the FAMOUS AOGCM, and Smith (2012, 10.5194/gmd-5-269-2012), which documents the most recent versions.

p3 line 12-14. See general comment (2), on nomenclature. Since this paper doesn't intend to survey all users and variants of HadCM3, it might be better to move the

statements in the latter part of this para, from line 17, to the start of the para, and thus begin by stating that BRIDGE has modified HadCM3 in various ways.

p3 line 29. At line 28 on p4, the authors explain that HadCM3-M1 is not exactly the model of Gordon et al. (2000). It would be useful to say that here, since I assumed that's what it was on first reading. See also general comment (2), on nomenclature.

p5 Table 1. Please give units for all quantities that aren't dimensionless. Some of these parameters are generally understandable, such as resolution and "no Iceland", but I suspect that many or most of them are not self-explanatory. While I appreciate the value in documentation, I think we have to consider what the purpose of the paper is. As far as most readers are concerned, the interest in these parameters will be their physical effect, and the consequences of changing them; if they are of sufficient interest to mention, these should be described in the text, with references to the literature of the schemes where appropriate. Writing down the values which appear in the configuration of these schemes is technical documentation, rather than scientific, so perhaps it's not needed in the paper, or maybe it would belong better in an appendix.

p6 line 8. This is not an accurate statement. The base code of UM4.5 is available at that URL, but the scientific definition of HadCM3 also depends on a lot of code mods. This is explained in Sect 7, but should be mentioned here too, or you could delete this statement and refer to Sect 7.

Sect 2.1-2.4. See general comment (1).

p14 line 32. See general comment (2) on nomenclature re HadCM3L.

p15 line 10. I'm not sure it's as simple as that. I think the introduction of the GM scheme is important in alleviating the need for flux adjustment.

p15 line 16 and Table 1. The coefficient being described is the vertical diffusivity - I would not call that a mixing rate. HadCM3 has a prescribed depth-dependent background vertical diffusivity and a Richardson-number-dependent part, which is important
near the surface. In FAMOUS, the latter part is omitted. The mixed-layer scheme is distinct from the vertical diffusion scheme.

p15 line 25-26. As far as I know, FAMOUS uses the same equation of state (UNESCO) and diffusion scheme as HadCM3 (except as noted in the last point).

p16 line 24-25. See general comment (2) on nomenclature re HadAM3H.

p16 line 4 and Table 1. I don't think the mention of RSOL will be intelligible, unless you explain what this means physically.

p19 line 26. Since it was earlier concluded that the two HadCM3s are the same, maybe it would be clearer to say "the CMIP5 historical experiment of HadCM3 done at the Hadley Centre", to avoid implying it's a significantly different version. Can you give a reference for this experiment?

p21 Fig 3a caption, same comment, "our version of HadCM3-M1" implies that it differs significantly from the Hadley Centre version.

p29 para from line 21. The net meridional freshwater transport may not be a reliable indicator of bistability. Sijp (Clim Dyn, 10.1007/s00382-011-1249-0, 2012) shows that it's the derivative wrt AMOC that may be the determining factor. Hawkins et al. (GRL, 2011, 10.1029/2011GL047208) demonstrate AMOC bistability in FAMOUS.

p30 line 3. Please give a citation to someone it is known by.

---

## Referee Comment (RC2) · Anonymous Referee #2 · 15 Mar 2017

general comments

=============

This paper describes the (largely shared) configurations of a family of climate models as used in a significant research group at Bristol University. The central family member, HadCM3, is well-known and was originally described in papers (and a number of non-peer reviewed technical reports, still readily available) published almost 20 years ago, although the other variants are much less well-described in the available literature, as far as I am aware. Despite the rather elderly nature of the models in question, they are still used and useful, and have been developed and modified in ways that I think sufficiently justifies revisiting and clarifying their documentation in the way the authors have done here. Treating HadCM3 and its spectrum of (roughly) resolution-

based variants as a family whose commonality and differences are best described together in one paper is a good approach, I feel.

Whilst it is impossible to cover every aspect of the performance of a global climate model, this paper covers a usefully illustrative spread of material, and is uniformly clear and well written. I have a few comments on specific areas, as detailed below, but on the whole I think the paper could be published in largely its current form.

My main general concern is based in the fact that this is a model description paper with little authorial connection to, or real acknowledgement of, the people who originally developed, coded and made the model available - presumably mostly Met Office and UGAMP/CGAM staff in the 1990s. As far as acknowledgement goes, I'm not sure how best this could be done, but it does feel like a very significant nod should be made in that direction. Content-wise, it would be good to be a little clearer on precisely what the BRIDGE authors have themselves added over the MetOffice/CGAM provided code and mods, especially considering that the full documentation for the Gordon et al HadCM3 is still available and that the HadCM3-M1 configuration described here is described as being almost identical to it.

specific comments

=============

WM="worth mentioning"

page 1, line 17: "version 1.0" - is the intention to upgrade the version number for the whole set of configurations any time there is a bug-fix or change that may only affect one of them? Putting the reference simulation ids in a more prominent place might help with the version/configuration tracking

p2,l10: "though" seems extraneous

p2,l17: "UKESM" the acronym should be explained, and a source given for this information. Perhaps "HadGEM3" is meant?

p2,l29: "Roberts et al. in review" - reference to (currently) unavailable literature

p3,l9: there are a number of other relevant model description papers for FAMOUS - the original Jones et al '05 paper should be additionally cited here at a minimum

p3,l14: Space needed between "Gordon et al.(2000)" and "HadCM3"

p3,l17: "this in relatively poorly documented" - perhaps "is"?

p4,l7: since a point is made of the computational speed of HadCM3 et al over more modern models, some detail of the computational throughput/resource requirements of the different configurations would be useful, either here or elsewhere

p4,l16,l17: "L" only refers to the ocean resolution, and "H" only to the atmosphere

p6,l15;p7,l29: WM - as a regular lat/lon grid is used in both components, Fourier filtering of higher wave-number dynamics is done in both models in certain latitude ranges.

p8,l6/p8,l25: WM - the rigid lid formulation requires the pre-specification of "islands" around which the barotropic circulation may occur. The standard MetOffice HadCM3, at least, does not allow mass transport through the Bering St because of this, which may affect the AMOC stability characteristics

p8,l30: WM - the virtual salinity flux is calculated using a globally constant reference salinity, which can distort the local response to the surface water forcing

p9,l30: despite what many generations of UM code and documentation has asserted, the soil hydrology in all versions of MOSES is apparently not derived from Clapp and Hornberger '78, but Brooks and Corey '64 (see eg footnote at http://jules-lsm.github.io/vn4.2/namelists/jules_soil.nml.html). The model still names everything with Clapp-Hornberger, so it would seem unhelpful to readers to start referring to Brooks and Corey when they won't find these names in the model, but this might be an opportunity to stop propagating the C-H misinformation.

p10,l4: I recall presentations by Valdes some years ago that appeared to show statistically different climates from the "same" HadCM3 ported to different platforms. Am I misremebering, or has this issue been cleared up to the authors' satisfaction?

p11,l4: MOSES2.2 can be used in FAMOUS, although most published FAMOUS papers use the MOSES1 variant.

p11,l21: WM - MOSES2.2 has two modes of operation. One functions as described here (calculating the exchange for each tile, then averaging the fluxes), and the other aggregates the different tile properties together /before/ doing one calculation of the average flux (see Essery et al 03). I assume the authors use the first mode - FAMOUS (eg the Williams et al '13 they cite) uses the second mode

p21-: from this point, some model names have an N or H appended (eg HadCM3-M2.2N) - I don't see where this is explained

p23,l12: Beware FAMOUS-M2.2! I believe new issues have very recently been found with the long-term drift of the climate of the model described in Williams et al (Smith, pers.comm) that may play into this sort of bias and require significant retuning. Do you know what the AMOC is doing in that run?

p25,l8: The section title is a little misleading, since this section is purely about heat transports. On the subject of pure TOA fluxes: I believe HadCM3 is known to have compensating biases in TOA short- and longwave fluxes, linked to known problems in the clouds. WM?

p27,l9: At shorter timescales, however, FAMOUS was found to have high levels of variability in the AMOC when compared to the RAPID data (Balan Sarojini et al, Ocean Science 2011)

p29,l17: the effectively shut Bering St may play a role here too

p29,l23-25: the end of this paragraph is phrased in rather too certain a manner concerning the reliability of the AMOC stability metrics presented eg in Liu et al'14 for my taste. It is too definite to claim that observations indicate that "the AMOC is in a

bistable regime". Theoretical metrics have been derived that suggest that this *may* be the case, but they are some way from being proved definitive.

p33,l1: Reading this section, the uninitiated might expect that they would be able to obtain and run the useful models described here themselves. In fact, they would have great difficulty even viewing the effective model code, given the web of libraries, patches and options (all, technically, supplied here) that the UM is built from. That is not the fault of the present authors, and neither is the fact that this well-known model effectively has no support or distribution mechanism. But I think that some warning should be given that the copious information supplied in this section is *not* really sufficient to run the model oneself, and perhaps contact details (maybe for Bryan Lawrence, Director of Models and Data at NCAS ) for someone to start with if a reader really did want to get help installing or running the system for their own use?

---

## Referee Comment (RC3) · C. Brierley (Referee) · 22 Mar 2017

**1   Overall Thoughts**

The version of HadCM3 used regularly within the group at Bristol has clearly branched from the Met. Office's original version. I think that this documentation (Valdes 2017) of it is a worthwhile contribution to GMD. I appreciate the provision of the source modifications and a list of the simulations. I would hope that in the final version there could be a link to the simulation output on the BRIDGE webpage. I had a couple queries, but suspect those could be addressed with revised sentences. I also found the figures and tables a little too small to be seen well on a printed version.

[Figure]

**2 Queries**

- Table 1 implies that two RHCrit values are due to level dependence rather than land/sea.

- Table 2 shows that MOSES2.2 (and hence TRIFFID) either is or cannot be used with the atmosphere only model. I wasn't sure why.

- What is the 3A spectral scheme (p7, L19)

- You state the soil layer thicknesses are a function of soil heat capacity and conductivity on p9, yet provide their depths on the p12. Are they constant across the globe?

- Are the surface types, LAI etc prescribed for each gridpoint (so 2D) or just each PFT.

- You state moving from MOSES2.1 to MOSES2.2 have "particularly big" effect. Can you either quantify or refer to later section.

- In section 4.1.3 and the Table 2, you state that there is a ratio of solar radiation components that is 0. Is this correct, and what does it mean physically.

- Section 4.1.5 seems redundant as only the sea-ice diffusivity in Table 1 hasn't been explicitly mentioned.

- What is the updating frequency of HadRM's lateral boundary conditions - you state 6 hours on p17 and 3-4 hours on p18.

- Please provide a little more information about the HadRM3 vs HadAM3 diffusion - there are 2 parameters in Table 1 and isn't clear to me what they mean.

- Some Figures (e.g. Fig2 ) appear to use different acronyms to the text.

- The discussion on p24 about leeward precipitation appears to be in the opposite sense as shown in figure 4.

- Section 5.1.3 does not discuss TOA flux - rather heat transports.

- Fig 5 contains no explanation of the gray lines.

- p27, l7. It isn't clear to me what is meant by "larger annual variation" - within year or between years.

- The references to Fig 9e/f need correcting

- The discussion in Section 5.3.2 doesn't necessarily recognise the big peak in the MODIS data at 25oN - rather it considers the feature symmetric aroudn the equator.

**3  Sentence Suggestions**

- p2, l2. I wonder if the final sentence of the abstract should use "predominantly" rather than "particularly".

- p2, l9. Flux corrections are not often discussed anymore, so I think you need to explain a little more

- p7, l3. "which determine" occurs twice in quick succession.

- p7, l13. Be explicit that this relates to RHCrit in Table 1.

- p7, l26. you may want to consider giving the origin of some of the default fields.

- p8, l14-15. This feels like duplication of prior sentence.

- p8, l24. "direct" -> "dynamic"?

- p9, l10. The end of this sentence reads awkwardly.

- p10, l6. Reference and explanation of the Visbeck scheme?

- p10, l11. "numerous" -> "multiple" and incorporate ref to supplementary info.

- p10, l19. wrong section reference

- p10, l24. This sentence has too many clauses. Split into two.

- p11, l21-24. This reads awkwardly

- p13, l16. Could this be related to section 3.1.2?

- p19, l16. Rename section to "surface temperature patterns"?

- p19, l32 " of 2" -> ", by 2"

- Fig2 caption. section symbol §in a variable name

- p25, l20. Spell FAMOUS correctly

- p26, l2. You may want to add that this is despite the OHT biases.

- p29, l15. remove paragraph break

- p30, l7. I was unsure what "this version" of HadCM3 was from previous sentence. Refer to figure panel

- p30. l11. Why only this resolution?

- p31, l10-12. Can you refer back to the definition of these different terms?

- p31, l24 "and which" -> ". We additionally show that"

---

## Author Comment (AC1) · 4 May 2017

**The purpose of this paper is to document the GCMs derived from HadCM3 that have been and will continue to be used by the authors for a wide range of climate research activities. This is an appropriate purpose for a GMD paper. The paper is clearly and well-written and serves the purpose. Its comparison of the results from various HadCM3-related configurations with each other and with CMIP5 models, to give an overview, is particularly useful. Although HadCM3 is quite an old model, it is still important. In the Web of Science, the paper of Gordon et al. (2000) on HadCM3 has been cited at least 100 times per year in the last decade. I have first some general comments, then some specific ones. I'm sure that my comments can be addressed, if the editor agrees that they should be, but since they're quite extensive I'm classifying them as implying major revisions.**

We thank the review for the time they have taken to improve the paper with their detailed comments.

**General comments**

**(1a) It's rather unusual for a paper to document a model that was mostly not developed by the authors, especially when other papers already exist about the model. Moreover the authors are not the sole users of this model. Of course the authors are aware of these points, and have cited papers other than theirs; I have suggested some more citations, and some minor rephrasing to avoid misinterpretation. In a way, this situation is like an open-source development (these models aren't open-source, although readily available).**

We do agree that the paper is unusual in that we have developed relatively small amount of the overall code, but it arose because reviewers were increasingly critical of the lack of documentation of the specific model versions that we were using in many other papers. In addition, reviewers were also critical of the overall skill of our models simply because it was old. Hence we wished to show that, at least for its mean climate, it remains competitive with the state-of-the-art models.

To better reflect some of these issues, we have invited Pope and Gordon to be co-authors and we am pleased to say they have accepted. Moreover, Michel Crucifix assembled the updates for MOSES2.1 and is also now an co-author.

**(1b) In view of this, the paper should focus on the aspects which BRIDGE specifically has contributed. For example, while the summaries of sects 2.1-2.4 of HadCM3 are clearly written, I feel that this much information is not needed, because it's essentially documentation of the original HadCM3, for which Pope et al., Gordon et al., and references therein should be cited. For the purpose of this paper, sect 2.5 is the important part, giving the differences of the BRIDGE HadCM3-M1 from the model of Gordon et al. These are stated to have very little effect.**

We have changed the tone to ensure there is no confusion about who wrote the model. The first sections are a brief summary of the original model for completeness. We feel that it does

help the reader understand the context of our paper, though we now strongly emphasis that the "new" aspects start at section 2.5.

**(2) The nomenclature of the models could be confusing. The purpose of this paper is to describe models used at BRIDGE, but these have to be clearly related to or distinguished from other models, in the same family, that have been used at the Met Office and elsewhere. HadCM3 and HadAM3 are models developed by the Hadley Centre.**

**On p3 line 12-14, the authors write, "Since its introduction, HadCM3 (and related models) has undergone a substantial number of changes, bug fixes and adaptations, such that few of the versions of the model used now are truly identical to their original model description in Pope et al. (2000) and Gordon et al. (2000)." I think this statement is debatable. It is likely that Hadley Centre users of HadCM3, and at least some other users, would say that it is not \*really\* HadCM3 if it's different from the model described by the papers of 2000, as far as the AOGCM is concerned, except for bug-fixes and mods needed for porting. That is, HadCM3 has not been revised. Of course it's hard to decide what's a bug-fix versus a scientific change - that's why this is debatable.**

**It turns out that the BRIDGE HadCM3-M1 is practically identical to the model of Gordon et al. (2000) (section 2.5). This is useful to know, but I'd suggest that it means it would be clearer to call this model just "HadCM3" throughout the paper.**

We agree that some of the models used by BRIDGE are very similar to those presented by the Met Office. We have further clarified this at a number of points in the text. However we believe that the changes made by BRIDGE should be defined and described. This removes ambiguity surrounding HadCM3 and the variants used at Bristol which is the key aim of the GMD journal (and has been asked for by reviewers of our papers).

Moreover, there is already some ambiguity in the use of the name HadCM3 throughout the community, particularly related to the land surface scheme. In one sense, the reviewer is correct to say that the core AOCGM "defines" HadCM3 but it is now common to use alternative land surface schemes and, in most cases, these are still referred to as HadCM3. For instance, it is difficult to determine if the HadCM3 featured in the model evaluation figure 9.1 in the IPCC AR5 report (corresponding to our figure 1) was the original model, or included MOSES2.1/TRIFFID or MOSES2.2/TRIFFID. Most of the text implied it was the original model but the model description table (9.A.1) references the Cox et al TRIFFID description paper (which was using MOSES2.1) and a Mercado et al paper which is MOSES2.2!

So we argue that it is important to improve the nomenclature for the model variants described here to more clearly distinguish the different versions as well as those developed at the Met Office. We have changed the nomenclature of the model acronyms to include the letter **B** for Bristol, so HadCM3-M2.1 becomes HadCM3B-M2.1 etc. We have continued to

use the HadCM3 acronym despite the differences in the model to that described by Gordon et al. and Pope et al., as the vast majority of the code remains the same.

To more clearly highlight what changes that have been made to the Bristol variants, we have moved the bug fixes section to section 2.1 before the descriptions of the components and split this section to define changes made to HadCM3-M1 and HadAM3.

In all cases we have retained the MOSES suffixes, as the name HadCM3 appears to frequently be used as a generic term regardless of the land surface scheme used. We think it would be a useful precedent to be more specific about which version of HadCM3 is being used to clearly distinguish this version from others using MOSES2.1 or 2.2.

**However, we learn that the BRIDGE HadCM3L is a very different model from the earlier Met Office model HadCM3L (p14 line 32). (A reference should be given for the Met Office HadCM3L - I think it might have been first used though not named by Cox et al., Nature, 2000). Similarly, the BRIDGE HadAM3H is said to be different from HadAM3H as used by the Met Office (p14 line 24-25). It is therefore confusing to call the Met Office and BRIDGE models by the same names.**

Agreed. We have now resolved this naming confusion by adopting the HadCM3B nomenclature throughout where the model departs from the Met Office model.

The Cox reference has now been added at the first mention of HadCM3L who as you say first uses the L flavour of the model (but does not call it HadCM3L).

**At least in the context of this paper, I suggest that models from BRIDGE and the Met Office are given different names, so that the reader can be clear, when differences from HadCM3 are mentioned, which are the ones introduced at BRIDGE, and which are the same as in the Met Office version. For HadRM3, is Hudson and Jones (a technical report on a particular application) the only reference that can be cited?**

The reference to Hudson and Jones was for HadAM3H. We have added a citation for HadRM3 too to help. We have also added a further reference for the use of HadAM3H.

**There are several published versions of FAMOUS. It would be useful if the authors could relate the BRIDGE version to those documented by Jones et al. (2005), Smith et al. (2008) and Smith (10.5194/gmd-5-269-2012, 2012). The Smith papers identify versions of FAMOUS by the run ID of the definitive UM basis file, so it would be useful to relate the runs detailed in Sect 7 to theirs. Also, FAMOUS is documented at www.famous.ac.uk, which could be cited.**

Our version of FAMOUS is identical to that described in the papers above. We have clarified this in our text. References to FAMOUS are only included to complete the set of models based on HadCM3. We have made that clearer in our text.

**(3) Effect of differences in configuration. It seems to me that there's not a very close connection between the model documentation of sect 2-4 and the results of sect 5. It would be valuable, wherever possible, to relate the differences in results to the differences in formulation. I appreciate that some of this is done, and that it's hard, but whatever more can be done would be useful, since it's the kind of information that would be relevant to deciding or understanding the choice of model for a particular purpose.**

We've added discussion in the temperature and land sections of section 5 about how the differences in sections 2-4 affect the performance of the model.

**(4) Computational cost (i.e. speed) is an important consideration in the choice of model. If one had infinite computing resource, for example, one would probably not prefer FAMOUS to HadCM3. It would be informative to collect some numbers for the relative computational cost of the model e.g. in Table 1.**

This is a good point raised by both reviewers. We have included a new table of run speeds for each of the MOSES1 models in typical configurations, run on the same machine (table 3).

**Specific comments**

**p1 line 8. I would delete "originally". Subsequent developments are smaller changes than the original development; HadCM3 is still essentially the same.**

We agree, this was misleading and has now been removed.

**p1 line 9-10. "but is now largely being replaced by more recent models" strikes me as an odd thing to say. It's true that HadCM3 is now little used by the Hadley Centre, which mostly uses later Met Office models. Some other centres, including BRIDGE, use HadCM3, but these centres, including BRIDGE, use other and later models too. Most centres, including the Hadley Centre, routinely use a range of models, older and newer, for different purposes. For reasons the paper explains, there are purposes for which older models are more suitable and cannot be replaced by later ones. I would suggest deleting this phrase.**

We would like to retain this phrase because HadCM3 has for many of the areas for which it was originally developed, been effectively completely superseded by more recently released models. These newer models either have higher resolution, include more complete physical/biological/chemical representations or have differences in the fundamental aspects, such as dynamics or clouds etc. We have thus rephrased this:
"... has been heavily used during the last 15 years for a range of future (and past) climate change studies but has now been largely superseded for many scientific studies by more recently developed models."

**p1 line 10. This paper is about the models used at BRIDGE, but other places use HadCM3 in the Met Office version. This sentence and the next two could seem to imply that BRIDGE is responsible for development of these models. To avoid that implication, you could slightly rephrase e.g. It continues to be used by various institutions, including by the BRIDGE (Bristol Research Initiative for the Dynamic Global Environment) research group at the University of Bristol, who have made adaptations over time to the base HadCM3 model. These adaptations mean that the original documentation is not entirely representative of the models used at BRIDGE, where several other configurations are in use which now differ from the originally described model versions.**

This is a very reasonable point and the sentence in question has been adjusted.

**p1 line 17. In the title and here you mention "version 1.0" for this suite of models, but you don't mention it again in the paper, which describes each of the configurations individually. Is it useful to assign a version number to the entire suite? If it is, it would be worth explaining what defines a version of the suite, and what would qualify as a new version.**

The GMD guidelines require that all model description papers have a version number. This leaves open the possibility of publishing updates in the future. When this version number is changed, and to what (e.g. v2.0, v1.1), will depend on the modification(s) being made and will be the topic of future discussions.

**p2 line 4. It would be worth spelling out that "model" is a general circulation model. Not all climate models are GCMs.**

We've added: "(all of which can be classed as General Circulation Models, GCMs)".

**p2 line 6. The statement is correct that the HadCM3 family is in use at BRIDGE, but it could seem to imply that BRIDGE is the only group which uses HadCM3 etc. still. As the authors know, and have stated at p3 line 14, it has other active users too.**

Entirely fair point, and we have modified the text to reflect that we are not the only users.

**p2 line 7. While it's true that Gordon et al. (2000) describe developments to the ocean model, I would say that main achievement and interest of that paper is HadCM3 i.e. the coupled model. Thus I would suggest phrasing this slightly differently e.g. "HadAM3 (Pope et al., 2000). Together with improvements to the ocean GCM, this enabled the development of HadCM3 (Gordon et al., 2000), which was one of the first coupled atmosphere-ocean GCMs that did not require ...".**

The text has been edited as per the reviewer suggestion.

**p2 line 9. Replace "was" with "has been", since it's still in use. It has been used for the past as well as the future. I think that at this point it would be appropriate to cite**

some of the major uses of HadCM3 that don't come up later, such as detection and attribution (Stott et al., Science, 2000), unforced variability (Collins et al., Clim Dyn, 2001), climate projection (Johns et al., Clim Dyn, 2003), uncertainty and constraints on projections (Stott and Kettleborough, Nature, 2002), decadal prediction (Smith et al., Science, 2007). I've made other suggestions later too.**

"Was" replaced. We have also inserted the suggested references.

**p2 line 10. Why is "though" at the end of the sentence?**

Apologies, it seems like the end of the sentence was cut off and it was somehow missed in the editing process. This has been corrected.

**p2 line 16. Reichler and Kim (BAMS, 2008) could be cited here as well.**

Added.

**p2 line 17. UKESM hasn't been introduced.**

Both reviewers pointed out the unexplained acronym. The text has been updated to say UK Met Office Unified Model (UM) rather than specifically UKESM.

**p2 line 24. Hewitt et al. (GRL, 2001) could be cited here as the first use of HadCM3 for a snapshot of palaeoclimate.**

Added.

**p2 line 26. Faster models are also useful for investigation of anthropogenic change on long timescales; HadCM3 has been used for that as well e.g. Gregory et al. (GRL, 2004), Ridley et al. (Clim Dyn, 2005).**

We have added this to the text and cited these examples.

**p2 line 25. Although multi-centennial rather than multi-millennial, I think it would be relevant to cite Tett et al. (Clim Dyn, 2007) as the first use of HadCM3 for a study of long-term past climate change. Among examples of more recent uses of HadCM3 for such a purpose, I would say that Schurer et al. (Nature Geosci, 2013) could be cited.**

We have added Tett et al., as an early multi-centennial example here.

**p2 line 27-29. What boundary conditions are meant here? I usually understand BCs as concerning forced climate change. Of course there are numerous earlier examples of climate-change experiments with HadCM3. Large initial-condition ensembles are a good application too, I agree, but I would say that a greater use of HadCM3 has been for parameter perturbation ensembles, the next point in this para, so maybe the order of these should be reversed.**

We have provided clarification of the meaning of the term boundary conditions and have reversed the order of typical usage of the models.

**p2 line 29-31. I'm unclear what distinction is being drawn between "multiple runs to explore the sensitivity" and "calculating probability density functions". Could these be merged? Murphy et al. (Nature, 2004, the first QUMP paper), and Stainforth et al. (Nature, 2005, the climateprediction.net paper) should be cited.**

These two have been merged together and Stainforth et al. (2005) cited.

**p3 line 9. Jones et al. (Clim Dyn, 2005) should be cited here as well, since it was the first publication of the FAMOUS AOGCM, and Smith (2012, 10.5194/gmd-5-269-2012), which documents the most recent versions.**

We have added the additional citations as requested.

**p3 line 12-14. See general comment (2), on nomenclature. Since this paper doesn't intend to survey all users and variants of HadCM3, it might be better to move the statements in the latter part of this para, from line 17, to the start of the para, and thus begin by stating that BRIDGE has modified HadCM3 in various ways.**

We have kept the structure but modified the emphasis that we are only describing BRIDGE usage. Moreover, we also correcting for the lack of documentation of MOSES2.1, and HAdCM3L which has been used by many groups not just BRIDGE.

**p3 line 29. At line 28 on p4, the authors explain that HadCM3-M1 is not exactly the model of Gordon et al. (2000). It would be useful to say that here, since I assumed that's what it was on first reading. See also general comment (2), on nomenclature.**

This has now been made clearer earlier in the text with the specific section to these modifications highlighted/linked.

**p5 Table 1. Please give units for all quantities that aren't dimensionless. Some of these parameters are generally understandable, such as resolution and "no Iceland", but I suspect that many or most of them are not self-explanatory. While I appreciate the value in documentation, I think we have to consider what the purpose of the paper is. As far as most readers are concerned, the interest in these parameters will be their physical effect, and the consequences of changing them; if they are of sufficient interest to mention, these should be described in the text, with references to the literature of the schemes where appropriate. Writing down the values which appear in the configuration of these schemes is technical documentation, rather than scientific, so perhaps it's not needed in the paper, or maybe it would belong better in an appendix.**

We have added the missing units to this table. We have elected to retain this table as it gives a broad easy reference comparison of the models in terms of resolution and inclusion or not of specific schemes (e.g. vertical ocean diffusion). Also the model parameter values given in this table are referred to in several places in the text, and are therefore integral to the detailed description of the different configurations that we are aiming for.

**p6 line 8. This is not an accurate statement. The base code of UM4.5 is available at that URL, but the scientific definition of HadCM3 also depends on a lot of code mods. This is explained in Sect 7, but should be mentioned here too, or you could delete this statement and refer to Sect 7.**

It has been clarified that extra modifications are required to the base code, along with a reference to sect. 7 where this is explained.

**p14 line 32. See general comment (2) on nomenclature re HadCM3L.**

This sentence has been moved to the first paragraph in Section 4.1 (second sentence), and the model name has been updated according to the reply to comment (2).

**p15 line 10. I'm not sure it's as simple as that. I think the introduction of the GM scheme is important in alleviating the need for flux adjustment.**

We have removed the reference to HadCM2, and instead reported the findings of Jones et al: "Jones et al (2003) investigated potential modifications to allow increased heat transport through this region, thus alleviating the unrealistic buildup of sea ice in the Nordic Sea, and concluded that the removal of Iceland was the preferred solution. With this modification, the improved meridional overturning circulation leads to more realistic heat transports in the coupled system and alleviates the need for flux correction."

**p15 line 16 and Table 1. The coefficient being described is the vertical diffusivity - I would not call that a mixing rate. HadCM3 has a prescribed depth-dependent background vertical diffusivity and a Richardson-number-dependent part, which is important near the surface. In FAMOUS, the latter part is omitted. The mixed-layer scheme is distinct from the vertical diffusion scheme.**

Table 1 has been updated to describe this coefficient as the "vertical tracer diffusivity" and the option used in BRIDGE versions of HadCM3L and FAMOUS is now described as "constant background value". The text of section 4.1.2. has been updated similarly.

**p15 line 25-26. As far as I know, FAMOUS uses the same equation of state (UNESCO) and diffusion scheme as HadCM3 (except as noted in the last point).**

We have clarified this in the text. FAMOUS and HadCM3 do use different approximation of the equations of states, but only within the diffusion scheme.

**p16 line 24-25. See general comment (2) on nomenclature re HadAM3H.**

Please see our previous responses on nomenclature.

**p16 line 4 and Table 1. I don't think the mention of RSOL will be intelligible, unless you explain what this means physically.**

This has been updated in the text to briefly explain this term.

**p19 line 26. Since it was earlier concluded that the two HadCM3s are the same, maybe it would be clearer to say "the CMIP5 historical experiment of HadCM3 done at the Hadley Centre", to avoid implying it's a significantly different version. Can you give a reference for this experiment?**

The text has been amended as suggested and references added.

**p21 Fig 3a caption, same comment, "our version of HadCM3-M1" implies that it differs significantly from the Hadley Centre version.**

Text modified.

**p29 para from line 21. The net meridional freshwater transport may not be a reliable indicator of bistability. Sijp (Clim Dyn, 10.1007/s00382-011-1249-0, 2012) shows that it's the derivative wrt AMOC that may be the determining factor. Hawkins et al. (GRL, 2011, 10.1029/2011GL047208) demonstrate AMOC bistability in FAMOUS.**

We have stated that freshwater transport may not be a reliable indicator and there remains uncertainty over this hypothesis, citing Sijp, (2012). We have also stated that FAMOUS has been shown to exhibit bistability and included the Hawkins et al. citation.

**p30 line 3. Please give a citation to someone it is known by.**

This understanding comes from personal communications. To avoid ambiguity, we have removed this to merely state what the models show here.

**general comments =============**

**This paper describes the (largely shared) configurations of a family of climate models as used in a significant research group at Bristol University. The central family member, HadCM3, is well-known and was originally described in papers (and a number of non-peer reviewed technical reports, still readily available) published almost 20 years ago, although the other variants are much less well-described in the available literature, as far as I am aware. Despite the rather elderly nature of the models in question, they are still used and useful, and have been developed and modified in ways that I think sufficiently justifies revisiting and clarifying their documentation in the way the authors have done here. Treating HadCM3 and its spectrum of (roughly) resolution based variants as a family whose commonality and differences are best described together in one paper is a good approach, I feel.**

**Whilst it is impossible to cover every aspect of the performance of a global climate model, this paper covers a usefully illustrative spread of material, and is uniformly clear and well written. I have a few comments on specific areas, as detailed below, but on the whole I think the paper could be published in largely its current form.**

We thank the reviewer for their kind comments and the time they have dedicated to helping us improve this paper.

**My main general concern is based in the fact that this is a model description paper with little authorial connection to, or real acknowledgement of, the people who originally developed, coded and made the model available - presumably mostly Met Office and UGAMP/CGAM staff in the 1990s. As far as acknowledgement goes, I'm not sure how best this could be done, but it does feel like a very significant nod should be made in that direction.**

As discussed in our response to reviewer 1, we accept that this is an unusual paper since we are effectively documenting small modifications to a large existing model that we did not develop. We have written the paper because a number of reviewers of our other papers have remarked about the lack of details for our modifications of the model. They have also opined that the model is not good enough (compared to CMIP5 models). We therefore decided that this paper is required.

We share your concerns about suitable credit to the original programmers, and therefore before we started writing the paper we contacted the Met Office, and they confirmed that they were happy with us writing the paper. However, we have addressed the reviewers concerns by inviting Vicky Pope and Chris Gordon to be co-authors. We have also invited Michel Crucifix (who brought together most of the HadCM3/MOSES2.1 version that is the centre of our work.

In response to your comments, we have revised throughout the paper to strengthen the fact that we are making modest modifications of a large existing code, and we have added to the

acknowledgements to thank everyone involved in the development of the model for their work.

**Content-wise, it would be good to be a little clearer on precisely what the BRIDGE authors have themselves added over the MetOffice/CGAM provided code and mods, especially considering that the full documentation for the Gordon et al HadCM3 is still available and that the HadCM3-M1 configuration described here is described as being almost identical to it.**

We've gone through the text and added stronger phrasing to clarify what are original model stuff and what are unique to BRIDGE.

We have also added a further comment with respect to HadCM3B-M1. The original UM distribution of the code had a line of code which had a serious compiler specific problem (identified by Dr. Lois Steenman-Clark, whom we acknowledge). On some compilers, it produced (statistically) the same results as shown by the Hadley centre. However, using alternative compilers the model was more than 0.75C cooler in the global mean. Some non-Hadley centre publications of HadCM3-M1 contained this bug. It is thus very important for users to be aware of this. We have added text to clarify this.

**specific comments =============**

**WM="worth mentioning"**

**page 1, line 17: "version 1.0" - is the intention to upgrade the version number for the whole set of configurations any time there is a bug-fix or change that may only affect one of them? Putting the reference simulation ids in a more prominent place might help with the version/configuration tracking**

The GMD guidelines require that all model description papers have a version number. This leaves open the possibility of publishing updates in the future. We are currently making some further modifications and whether these represent new version numbers will be decided when we publish.

**p2,l10: "though" seems extraneous**

We have removed the word "though" from the end of the sentence.

**P2, l17: "UKESM" the acronym should be explained, and a source given for this information. Perhaps "HadGEM3" is meant?**

Both reviewers pointed out the unexplained acronym. The text has been updated to say UK Met Office Unified Model (UM) rather than specifically UKESM.

**p2,l29: "Roberts et al. in review" - reference to (currently) unavailable literature**

This reference has been deleted.

**P3, l9: there are a number of other relevant model description papers for FAMOUS - the original Jones et al '05 paper should be additionally cited here at a minimum**

This reference and Smith 2012 have been added.

**p3,l14: Space needed between "Gordon et al.(2000)" and "HadCM3" p3,l17: "this in relatively poorly documented" - perhaps "is"?**

Thank you for picking this up, it has now been correct.

**P4, l7: since a point is made of the computational speed of HadCM3 et al over more modern models, some detail of the computational throughput/resource requirements of the different configurations would be useful, either here or elsewhere**

This is a good point raised by both reviewers. We have included a new table of run speeds (table 3).

**p4,l16,l17: "L" only refers to the ocean resolution, and "H" only to the atmosphere**

This is correct and has been clarified in the text.

**p6,l15;p7,l29: WM - as a regular lat/lon grid is used in both components, Fourier filtering of higher wave-number dynamics is done in both models in certain latitude ranges.**

A sentence has been added to each of these sections (atmosphere and ocean) to state this.

**p8,l6/p8,l25: WM - the rigid lid formulation requires the pre-specification of "islands" around which the barotropic circulation may occur. The standard MetOffice HadCM3, at least, does not allow mass transport through the Bering St because of this, which may affect the AMOC stability characteristics.**

The formulation of islands in the model is discussed in Section 4.1.4. Here, we add: "The barotropic solver requires the pre-specification of "islands" around which the barotropic circulation may occur (See Section 4.1.4)"

(Some of our current work on development is allowing mass transport through the Bering Strait and it does indeed have a big effect on the AMOC).

**p8,l30: WM - the virtual salinity flux is calculated using a globally constant reference salinity, which can distort the local response to the surface water forcing**

Thank you for pointing this out, we've added this sentence to the paper.

**p9,l30: despite what many generations of UM code and documentation has asserted, the soil hydrology in all versions of MOSES is apparently not derived from Clapp and Hornberger '78, but Brooks and Corey '64 (see eg footnote at http://juleslsm.github.io/vn4.2/namelists/jules_soil.nml.html). The model still names everything with Clapp-Hornberger, so it would seem unhelpful to readers to start referring to Brooks and Corey when they won't find these names in the model, but this might be an opportunity to stop propagating the C-H misinformation.**

Interesting. I was not aware of this history. A quick reading of the relevant papers suggests that Brooks and Corey devised the expression but C-H calculated more values. We have added a sentence and reference to Brooks and Corey.

**p10,l4: I recall presentations by Valdes some years ago that appeared to show statistically different climates from the "same" HadCM3 ported to different platforms. Am I misremebering, or has this issue been cleared up to the authors' satisfaction?**

Well remembered. This problem has indeed been sorted. There were two problems. One was the compiler bug (mentioned above) and the second was related to a rather strange issue related to reconfiguration. We now get the same climate (statistically) for HadCM3-M1 and our PMIP2 simulations are the same too.

**p11,l4: MOSES2.2 can be used in FAMOUS, although most published FAMOUS papers use the MOSES1 variant.**

We've added a comment about this to the text.

**p11,l21: WM - MOSES2.2 has two modes of operation. One functions as described here (calculating the exchange for each tile, then averaging the fluxes), and the other aggregates the different tile properties together /before/ doing one calculation of the average flux (see Essery et al 03). I assume the authors use the first mode - FAMOUS (eg the Williams et al '13 they cite) uses the second mode**

This is correct, we use the first mode. Essery et al. 2003 note that there are negligible differences in the result from the two approaches but that conceptually the first mode is easier within a GCM.

**p21-: from this point, some model names have an N or H appended (eg HadCM3-M2.2N) - I don't see where this is explained**

Corrected.

**p23,l12: Beware FAMOUS-M2.2! I believe new issues have very recently been found with the long-term drift of the climate of the model described in Williams et al (Smith, pers.comm) that may play into this sort of bias and require significant retuning. Do you know what the AMOC is doing in that run?**

We have performed a 1000 year run and the surface temperature and AMOC do not appear to have a drift (though the AMOC does have a lot of variability, see below) but we are aware of the issues that the reviewer alludes to. However, since these are still being investigated and FAMOUS is largely documented elsewhere, we have not amended the text.

**p25,l8: The section title is a little misleading, since this section is purely about heat transports. On the subject of pure TOA fluxes: I believe HadCM3 is known to have compensating biases in TOA short- and longwave fluxes, linked to known problems in the clouds. WM?**

This section has been renamed "Horizontal heat transports".

**p27,l9: At shorter timescales, however, FAMOUS was found to have high levels of variability in the AMOC when compared to the RAPID data (Balan Sarojini et al, Ocean Science 2011)**

We have added this to the end of this sentence and cited Balan Sarojini et al.

**p29,l17: the effectively shut Bering St may play a role here too**

Pardaens et al (2003) ran a sensitivity study allowing flow equivalent to an open Bering strait and found that it had little impact on the salinity error but did have a big impact on the AMOC. We are actively working on improving the Bering St and, if satisfactory, will probably be an update to our version.

**p29,l23-25: the end of this paragraph is phrased in rather too certain a manner concerning the reliability of the AMOC stability metrics presented eg in Liu et al'14 for my taste. It is too definite to claim that observations indicate that "the AMOC is in a bistable regime". Theoretical metrics have been derived that suggest that this \*may\* be the case, but they are some way from being proved definitive.**

We have rephrased the sentence to lower the emphasis placed on the certainty of the "observations"

**p33,l1: Reading this section, the uninitiated might expect that they would be able to obtain and run the useful models described here themselves. In fact, they would have great difficulty even viewing the effective model code, given the web of libraries, patches and options (all, technically, supplied here) that the UM is built from. That is not the fault of the present authors, and neither is the fact that this well-known model effectively has no support or distribution mechanism. But I think that some warning should be given that the copious information supplied in this section is \*not\* really sufficient to run the model oneself, and perhaps contact details (maybe for Bryan Lawrence, Director of Models and Data at NCAS ) for someone to start with if a reader really did want to get help installing or running the system for their own use?**

The text has been updated to make it clear that the code can only be viewed at the link given and to direct the reader to the UM Partnership Team for enquiries about using the model.

**The version of HadCM3 used regularly within the group at Bristol has clearly branched from the Met. Office's original version. I think that this documentation (Valdes 2017) of it is a worthwhile contribution to GMD. I appreciate the provision of the source modifications and a list of the simulations. I would hope that in the final version there could be a link to the simulation output on the BRIDGE webpage. I had a couple queries, but suspect those could be addressed with revised sentences. I also found the figures and tables a little too small to be seen well on a printed version.**

Thank you to Chris for his insightful comments and support of this paper.

**2 Queries**

**Table 1 implies that two RHCrit values are due to level dependence rather than land/sea.**

This is correct, the values of RHCrit in Table 1 do vary with level.

**Table 2 shows that MOSES2.2 (and hence TRIFFID) either is or cannot be used with the atmosphere only model. I wasn't sure why.**

We have changed the table to clarify that M2.2 *could* be used with HadAM3 and HadRM3, however it has not been done at Bristol (though would be simple to implement).

**What is the 3A spectral scheme (p7, L19)**

Spectral scheme 3A is a radiation scheme developed by the Met Office for version HadCM3 of the UM which is designed to treat both long- and shortwave schemes in a common, flexible framework as far as possible and is documented in Ingram, Woodward and Edwards (1997; Unified Model Documentation Paper No. 23: Radiation, http://cms.ncas.ac.uk/documents/vn4.5/p023.pdf). We have changed how we introduce this term and improved the referencing to make it clear where this term originates.

**You state the soil layer thicknesses are a function of soil heat capacity and conductivity on p9, yet provide their depths on the p12. Are they constant across the globe?**

The sentence written in the original manuscript has confused the reviewer: it is the amount of frozen soil moisture that is a function of soil heat capacity and conductivity, not the soil layer thickness. We have rewritten this sentence to make this clearer.

**Are the surface types, LAI etc prescribed for each gridpoint (so 2D) or just each PFT.**

The fractions of surface types and and values of LAI, canopy height and canopy conductance are specified at all grid points. The vegetation fractions and these other vegetation parameters will be updated by TRIFFID if used. Other parameters are hard-wired in the code. The text has been updated to clarify this.

**You state moving from MOSES2.1 to MOSES2.2 have "particularly big" effect. Can you either quantify or refer to later section.**

On reflection, we agree that 'particularly big' isn't a helpful quantification, so we've changed this phrase to merely state that it affects the temperature.

**In section 4.1.3 and the Table 2, you state that there is a ratio of solar radiation components that is 0. Is this correct, and what does it mean physically.**

An explanation of this has been added to section 4.1.4 and table 1. It basically controls the way that the solar flux penetrates into the ocean and has been renamed to solar penetrative flux.

**Section 4.1.5 seems redundant as only the sea-ice diffusivity in Table 1 hasn't been explicitly mentioned.**

We agree and have restructured the section and removed the mention of this.

**What is the updating frequency of HadRM's lateral boundary conditions - you state 6 hours on p17 and 3-4 hours on p18.**

This can vary between 3 - 6 hours. In these simulations we used 4-hourly updating.

**Please provide a little more information about the HadRM3 vs HadAM3 diffusion - there are 2 parameters in Table 1 and isn't clear to me what they mean.**

An explanation of these terms has been added to section 2.2.

**Some Figures (e.g. Fig2 ) appear to use different acronyms to the text.**

Thank you for pointing this out. The heading on Figure 2 has been updated to match the caption. The other figures have also been checked for consistency.

**The discussion on p24 about leeward precipitation appears to be in the opposite sense as shown in figure 4.**

We have clarified this section to highlight that we are discussing the bias in the gauge stations themselves rather than the models.

**Section 5.1.3 does not discuss TOA flux - rather heat transports.**

We have amended section title to "Heat transport"

**Fig 5 contains no explanation of the gray lines.**

We have added into the figure caption that these lines represent some of the CMIP5 models.

**p27, l7. It isn't clear to me what is meant by "larger annual variation" - within year or between years.**

We have clarified these sentences as follows:
"HadCM3BL shows larger year to year variability than the observations: approximately twice as large as that in the observations. This results in years with a lower minimum volume transport than are seen in the observations. FAMOUS model variants tend to underestimate

the year to year variation by approximately (50%), although this is in contrast to the study of Sarojini 2011 who showed that FAMOUS exhibited greater short-term variability than the RAPID-MOCHA array. HadCM3B variants have a realistic year to year variability at least in the upper (1500 m) of the ocean."

**The references to Fig 9e/f need correcting**
Apologies, these were left over from a previous version and have now been amended.

**The discussion in Section 5.3.2 doesn't necessarily recognise the big peak in the MODIS data at 25oN - rather it considers the feature symmetric around the Equator.**

We did refer to the 'subtropical spikes in productivity', but we have rephrased this to emphasise this spike at ~20N.

**3 Sentence Suggestions**

**p2, l2. I wonder if the final sentence of the abstract should use "predominantly"**

**rather than "particularly".**

Changed.

**p2, l9. Flux corrections are not often discussed anymore, so I think you need to**

**explain a little more**

We have added a short sentence here to clarify the role of flux adjustments.

**p7, l3. "which determine" occurs twice in quick succession.**

This has been rephrased.

**p7, l13. Be explicit that this relates to RHCrit in Table 1.**

We have added RHCrit to the sentence.

**p7, l26. you may want to consider giving the origin of some of the default fields.**

We have expanded the text accordingly.

**p8, l14-15. This feels like duplication of prior sentence.**

Deleted.

**p8, l24. "direct" -> "dynamic"?**

Corrected.

**p9, l10. The end of this sentence reads awkwardly.**

This has now been modified.

**p10, l6. Reference and explanation of the Visbeck scheme?**

We added a reference a brief outline of the purpose of the 'Visbeck' scheme.

**p10, l11. "numerous" -> "multiple" and incorporate ref to supplementary info.**

Word changed and reference added to the supp. Information at the end of the list.

**p10, l19. wrong section reference**

This has been corrected.

**p10, l24. This sentence has too many clauses. Split into two.**

Done.

**p11, l21-24. This reads awkwardly**

Repharased.

**p13, l16. Could this be related to section 3.1.2?**

There is a partial link, which we have added to the text.

**p19, l16. Rename section to "surface temperature patterns"?**

Done.

**p19, l32 " of 2" -> ", by 2"**

Done.

**Fig2 caption. section symbol §in a variable name**

This has been corrected.

**p25, l20. Spell FAMOUS correctly**

Corrected.

**p26, l2. You may want to add that this is despite the OHT biases.**

We have added this statement to the end of the sentence.

**p29, l15. remove paragraph break**

Done.

**p30, l7. I was unsure what "this version" of HadCM3 was from previous sentence. Refer to figure panel**

We have amended the sentence to make reference to the HadCM3 models shown in Figure 9.

**p30. l11. Why only this resolution?**

The incorrect modelling of C4 grasses at the mouth of the Amazon by HadCM3 is related to the precipitation bias seen in Fig 4. The text has been updated to make this connection more clearly.

**p31, l10-12. Can you refer back to the definition of these different terms?**

Done.

**p31, l24 "and which" -> ". We additionally show that"**

Done.

[revised manuscript text omitted]

---

## Author Response (AR2)

**Response to the Topical Editor**

**Thank you for submitting the revise manuscript, and detailed response.**

**I am convinced that almost all of the reviewers' comments are addressed comprehensively. The increased recognition of the extensive underlying model development undertaken elsewhere, and the justification for the need for this manuscript highlighted in your response to reviewer comment 1a, being the most important changes in my eyes.**

Thank you very much for your comments and the recognition of our work.

**Before this manuscript is accepted for publication, please can you address two further comments.**

Please see the response to each of your comments below. All page and line numbers refer to the LaTeXdiff version of the manuscript, appended to this response, which is a comparison of this submission with our previous submission.

**1) Please address (either by following the suggestion or explaining why you disagree), the as far as I can see not addressed high-level comment by Reviewer 3 'I would hope that in the final version there could be a link to the simulation output on the BRIDGE webpage'.**

We have added a link to the BRIDGE website which contains all the data for all the experiments we have used in the paper on lines 8–9, page 36. They can be found at `http://www.paleo.bristol.ac.uk/ummodel/scripts/papers/Valdes_et_al_2017.html`

**2) The addition of Table 3 is most welcome, but as far as I can see it is not referred to in the manuscript. In addressing this I suggest that you also highlight that the model speed is unlikely to scale linearly with the number of cores. This will avoid any confusion by non-specialists, and acknowledge this limitation in the presented information.**

Table 3 had been referred to in Section 1.1 on page 4. However we acknowledge the need to mention the non linearity of the relationship between the number/speed of CPU cores and the efficiency of the model. We have added a note to that effect in that paragraph (lines 26–29).

**Thanks again for your extensive and considered revisions.**

Thank you.

[revised manuscript text omitted]